# Integrated genomic analyses of *de novo* pathways underlying atypical meningiomas

Akdes Serin Harmancı[1,2], Mark W. Youngblood[1,2,3], Victoria E. Clark[1,2,3], Süleyman Coşkun[1,2], Octavian Henegariu[1,2,3,4,5], Daniel Duran[1,2], E. Zeynep Erson-Omay[1,2], Leon D. Kaulen[1,2], Tong Ihn Lee[6], Brian J. Abraham[6], Matthias Simon[7], Boris Krischek[8], Marco Timmer[8], Roland Goldbrunner[8], S. Bülent Omay[1,2], Jacob Baranoski[1,2,3], Burçin Baran[1,2], Geneive Carrión-Grant[1,2], Hanwen Bai[1,3], Ketu Mishra-Gorur[1,2,3,4,5], Johannes Schramm[7], Jennifer Moliterno[1,2], Alexander O. Vortmeyer[9], Kaya Bilgüvar[1,3,10], Katsuhito Yasuno[1,2], Richard A. Young[6,11] & Murat Günel[1,2,3,4,5,12]

Meningiomas are mostly benign brain tumours, with a potential for becoming atypical or malignant. On the basis of comprehensive genomic, transcriptomic and epigenomic analyses, we compared benign meningiomas to atypical ones. Here, we show that the majority of primary (*de novo*) atypical meningiomas display loss of *NF2*, which co-occurs either with genomic instability or recurrent *SMARCB1* mutations. These tumours harbour increased H3K27me3 signal and a hypermethylated phenotype, mainly occupying the polycomb repressive complex 2 (PRC2) binding sites in human embryonic stem cells, thereby phenocopying a more primitive cellular state. Consistent with this observation, atypical meningiomas exhibit upregulation of EZH2, the catalytic subunit of the PRC2 complex, as well as the E2F2 and FOXM1 transcriptional networks. Importantly, these primary atypical meningiomas do not harbour TERT promoter mutations, which have been reported in atypical tumours that progressed from benign ones. Our results establish the genomic landscape of primary atypical meningiomas and potential therapeutic targets.

[1] Yale Program in Brain Tumor Research, Yale School of Medicine, New Haven, Connecticut 06510, USA. [2] Department of Neurosurgery, Yale School of Medicine, New Haven, Connecticut 06510, USA. [3] Department of Genetics, Yale School of Medicine, New Haven, Connecticut 06510, USA. [4] Department of Neurobiology, Yale School of Medicine, New Haven, Connecticut 06510, USA. [5] Yale Program on Neurogenetics, Yale School of Medicine, New Haven, Connecticut 06510, USA. [6] Whitehead Institute for Biomedical Research, 9 Cambridge Center, Cambridge, Massachusetts 02142, USA. [7] Department of Neurosurgery, University of Bonn Medical School, Bonn 53105, Germany. [8] Department of General Neurosurgery, University Hospital of Cologne, Cologne 50937, Germany. [9] Department of Pathology, Yale School of Medicine, New Haven, Connecticut 06510, USA. [10] Yale Center for Genome Analysis, Yale School of Medicine, Orange, Connecticut 06477, USA. [11] Department of Biology, Massachusetts Institute of Technology, Cambridge, Massachusetts 02139, USA. [12] Yale Comprehensive Cancer Center, Yale School of Medicine, New Haven, Connecticut 06510, USA. Correspondence and requests for materials should be addressed to M.G. (email: murat.gunel@yale.edu).

Over 35% of all primary tumours that affect the central nervous system are meningiomas, which originate from the three-layer meningeal membrane ensheathing the brain and spinal cord[1]. According to the World Health Organization (WHO) criteria, meningiomas are classified into three pathological grades, based mainly on morphologic findings[2,3]. These include histological criteria such as mitotic activity, cellularity, cellular morphology and growth pattern, necrosis, and brain invasion. Approximately 70–80% of meningiomas are grade I and benign, while grade II and III meningiomas are higher grade and classified as atypical (5–20%) or malignant (1–3%), respectively[2,3]. The grading of a tumour carries prognostic value, as higher grade lesions are more likely to recur and decrease the chances of long-term survival[4].

Work by our lab and others has identified mutually exclusive molecular subgroups in benign meningioma, including loss of NF2 (occasionally with recurrent mutations in SMARCB1), mutations in the WD40 repeat region of TRAF7 (co-occurring with either PI3K activating mutations or recurrent KLF p.Lys409Gln mutation), activation of Hedgehog signalling (via SMO, SUFU or PRKAR1A), and recurrent p.Gln403Lys and p.Leu438_His439del mutations in the dock domain of POLR2A[5–7]. Interestingly, these subgroups were associated with distinct pathological and clinical findings. For example, secretory meningiomas were driven exclusively by TRAF7/KLF4 co-mutations, while fibrous meningiomas were primarily associated with NF2 loss. The intracranial origin of a meningioma was also predicted by the underlying meningioma mutations, with non-NF2 mutant tumours being enriched in the neural crest derived anterior skull base region, while samples harbouring NF2 loss arose from the mesoderm-derived posterior regions[6].

While these studies led to comprehensive genomic characterization of benign meningiomas, the genomic pathways that lead to formation of atypical cases are not well established. Primary atypical meningiomas form de novo, whereas secondary ones form due to recurrence and malignant progression of benign meningiomas[8]. Overall, atypical tumours comprise ~5–20% of all meningiomas and are associated with poor prognosis and a 10-year survival less than 80%, mostly due to local recurrence[9]. As with other high-grade forms of neoplasia, these tumours show increased genomic instability, including loss of chromosomes 22q, 1p and 14q[6,10]. Other than identifying these large chromosomal events, studies to date have failed to identify specific molecular changes that distinguish atypical from benign meningiomas.

The genomic alterations that differentiate low-grade and high-grade tumours have been extensively studied in other forms of neoplasia[11]. While somatic coding mutations have been identified in many cases, epigenetic modifications have also emerged as a potent mechanism to induce formation of malignant tumours. These include alterations in histone modifications or DNA methylation, both of which remodel chromatin to affect changes in gene expression and alter the transcriptional profile of the cancer cells[11,12].

Here, we used integrated genomic techniques to dissect the molecular landscape of primary atypical tumours compared with their benign counterparts. We show that the mutational background of primary atypical meningiomas to be comprised mainly of NF2 mutants, which frequently co-occur with either chromosomal instability or recurrent p.(Arg383Gln) or p.Arg386His mutations in SMARCB1 (co-occurrence $P = 1.2 \times 10^{-7}$, Fisher's exact test). Meningiomas with these alterations carry a significantly higher risk of being atypical as compared with non-NF2 meningiomas, including TRAF7 (with PI3K or KLF4 alterations), Hedgehog or POLR2A mutant tumours. Differences in the number of coding mutations do not contribute to the risk of being atypical; while large-scale copy number variant (CNV) events, transcriptional and epigenetic changes as well as alterations in miRNAs show substantial association. Indeed, genomically unstable, hypermethylated NF2 mutant meningiomas, which display activation of the cell cycle as well as the PRC2 pathways, account for the majority of the genomic landscape of primary atypical meningiomas. These findings define novel therapeutic targets in atypical meningiomas, which continue to represent significant treatment challenges due to a lack of effective chemotherapeutics.

## Results

**NF2 and SMARCB1 mutations in atypical meningiomas.** We hypothesized that similar to gliomas, in which malignancy (glioblastoma multiforme) can occur either de novo or through progression of a lower grade tumour[11], the molecular pathways that underlie formation of primary atypical meningiomas would be different than those associated with the progression of benign tumours[8]. On the basis of this assumption, we choose to specifically focus our analysis on primary atypical samples, as the molecular features of this tumour have not been described. Of the 775 meningiomas that we studied using next-generation exome ($n = 107$) or targeted sequencing, we concentrated our initial efforts on comparing histologically benign samples ($n = 468$) with de novo atypical tumours ($n = 88$) (Supplementary Fig. 1; Supplementary Data 1a–c). This dataset did not include tumours that have undergone previous chemotherapy or radiation, as these treatments may induce exogenous genomic changes.

We first searched for known meningioma driver mutations in each sample, defining the distribution of benign versus atypical meningiomas in each of the molecular subgroups including NF2, TRAF7 (co-mutated with PI3K pathway or KLF4), Hedgehog and POLR2A mutant tumours. We found significant differences in the percentage of atypical versus benign tumours within these molecular subgroups. Of the 88 histologically atypical meningiomas, 75% contained NF2 mutations (Supplementary Data 1c), while the remaining 9% were TRAF7/PI3K mutant and 16% did not harbour a mutation in the previously established meningioma genes. In our sample set, we did not observe any atypical tumours that harboured mutations in TRAF7/KLF4, POLR2A or the Hedgehog pathway. Because of the high prevalence of NF2 alterations in the atypical cohort, a tumour harbouring NF2 loss has a 3.78 times greater risk to be atypical compared with a non-NF2 meningioma ($P = 2.2 \times 10^{-10}$, Fisher's exact test).

Given these results, which suggest that primary atypical meningiomas were overwhelmingly associated with loss of NF2, we divided our cohort into two large subgroups: those with NF2 mutations, and those that were NF2 wild type (including TRAF7/PI3K//KLF4, Hedgehog and POLR2A mutant tumours as well as mutation unknown samples). We investigated the potential role of coding variation by comparing the number of somatic coding mutations in atypical and benign samples stratified by NF2 status, which allowed us to control for the underlying driver mutation. We did not find a statistically significant difference (Student's t-test) (Fig. 1a,b; Supplementary Figs 2 and 3; Supplementary Data 2a–e).

Among all samples that underwent whole-exome sequencing, the only somatically mutated gene that we found to be enriched in the atypical samples was NF2 (Supplementary Data 2b–e). We next studied an extended cohort that included not only the whole-exome sequenced dataset, but also an independent dataset of meningioma samples that underwent targeted sequencing ($n = 556$). We calculated the significance of association of the

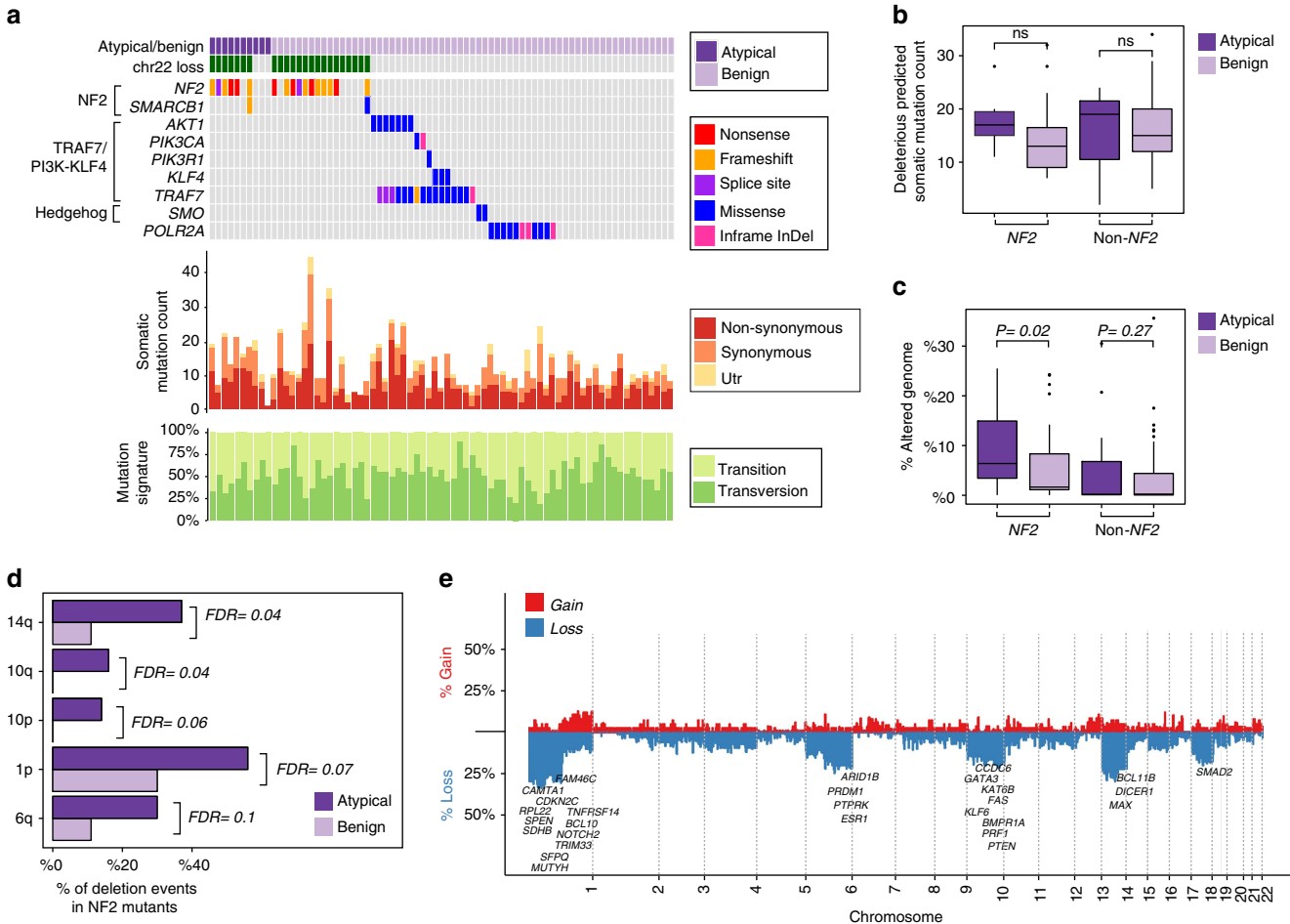

**Figure 1 | Somatic mutations and copy number variations in benign and atypical meningiomas.** (**a**) Oncoprint depicting the mutational profile of 75 exome-sequenced meningioma samples is shown. Histological grade, chromosome 22 loss status and recurrently mutated genes, which are clustered based on mutually exclusive meningioma subgroups (left) are summarized at the top panel. The distributions of somatic mutations according to their functional consequences are shown in the middle panel, whereas the mutational signatures are shown at the bottom. The colour codes are explained on the right. (**b**) Atypical versus benign meningiomas harbour similar number of damaging somatic mutations (atypical $n = 10$, benign $n = 65$). The lines above the bars indicate statistical analysis (Student's $t$-test; ns: non-significant). Lines depict the median values; boxes plot 25th to 75th percentiles, whereas separately plotted dots show the outliers. (**c**) The percentage of genome alteration is statistically significantly different between atypical versus benign meningiomas (atypical $n = 55$, benign $n = 153$) (Student's $t$-test). Lines depict the median values; boxes plot 25th to 75th percentiles, whereas separately plotted dots show the outliers. (**d**) Large-scale genomic events associated with atypical $NF2$ samples are shown (atypical $n = 43$, benign $n = 57$). Losses of chromosomes 14q, 10q, 10p, 1p and 6q significantly associate with atypical tumours (FDR adjusted Fisher's exact test). (**e**) Differences in genomic alterations between atypical $NF2$ and benign $NF2$ mutant samples are shown (atypical $n = 43$, benign $n = 57$). Along the horizontal axis, losses are depicted in blue, whereas gains are shown in red. The vertical axis represents the genome. Significantly altered tumour suppressor consensus cancer genes in atypical $NF2$ samples are noted.

driver mutations ($NF2$, $SMARCB1$, $TRAF7/PI3K$, $TRAF7/KLF4$, $POLR2A$, Hedgehog) with atypical meningiomas. We observed a significant association of being atypical only with $NF2$ and $SMARCB1$ mutations (NF2: $P = 2.2 \times 10^{-10}$, $SMARCB1$: $P = 0.05$, Fisher's exact test). These findings suggest that except for mutations in $NF2$ and $SMARCB1$, coding variation does not significantly contribute to the risk of being atypical.

We next calculated the clonality rate of each somatic mutation based on the variant allele frequency and ploidy at that site, considering the admixture rate of each tumour. In the majority of our samples (13/17), we calculated $NF2$ mutations to have a clonality rate near 100%, suggesting that $NF2$ mutations occurred early during tumour formation (Supplementary Data 2f; Supplementary Fig. 4).

**Atypical $NF2$ mutants demonstrate chromosomal instability.** Using whole-genome genotyping (WGG), we next compared

large-scale CNV events between benign and atypical tumours ($n = 153$ versus 55, benign versus atypical, respectively) (Supplementary Data 3a). We defined 'large-scale' CNV events as affecting more than one-third of a chromosomal arm and calculated the percentage of genome alteration (PGA) as a general indicator of genomic aberrations. Considering a calculated mean PGA value of 4.8% across 208 meningiomas, we classified any meningioma with a higher or lower PGA value as 'CNV-high' or 'CNV-low', respectively (Supplementary Fig. 5).

Overall, primary atypical $NF2$ tumours were significantly more likely to be CNV-high, as compared with benign $NF2$ ones ($P = 0.02$, Student's $t$-test, Fig. 1c). On the basis of correction for mutational background by considering only the $NF2$ mutant meningiomas, we calculated atypical tumours to have a 2.19 times greater risk to be CNV-high ($P = 0.001$, Fisher's exact test) (Supplementary Data 3b). When we performed the same analysis considering non-$NF2$ mutant meningiomas ($n = 96$ versus 12,

benign versus atypical, respectively), we did not find any statistically significant difference in PGA, likely due to the small number of non-*NF2* mutant meningiomas ($P = 0.27$, Student's *t*-test, Fig. 1c). Not surprisingly, when we combined the benign and atypical samples together and simply compared non-*NF2* mutants to *NF2* mutant meningiomas (among which percentage of atypical tumours is much higher), we obtained a similar result, with *NF2* mutants being genomically more unstable. These findings demonstrate that atypical tumours with *NF2* loss display increased chromosomal instability compared with their benign counterparts, while those without *NF2* loss do not.

We next compared specific CNV events associated with atypical samples, initially considering the *NF2* mutants only. As expected, all benign as well as atypical *NF2* mutant samples demonstrated chromosome 22 loss. There were, however, significant differences with respect to several other chromosomes. We found that chromosome 14q loss was the most significant difference between atypical and benign *NF2* meningiomas (18% versus 7%, respectively; $P = 0.0016$ false discovery rate (FDR) $= 0.04$, Fisher's exact test). This was followed by whole arm losses of chromosomes 10q, 10p, 1p and 6q, ($P = 0.002$ FDR $= 0.04$, $P = 0.005$ FDR $= 0.06$, $P = 0.008$ FDR $= 0.07$, $P = 0.013$ FDR $= 0.1$ respectively, Fisher's exact test) (Fig. 1d). No CNV events were significantly associated with atypical non-*NF2* meningiomas compared with benign non-*NF2* samples.

We next identified tumour suppressor genes located within these genomic regions frequently deleted in atypical *NF2* meningiomas. Deletions in phosphatase and tensin homologue (*PTEN*) (10q23) (13% versus 0% in atypical versus benign *NF2* tumours, FDR $= 0.018$), myc-associated factor X (*MAX*) (14q23) (34% versus 10%, FDR $= 0.03$), neurogenic locus notch homolog protein 2 (*NOTCH2*) (1p13) (20% versus %3, FDR $= 0.043$), AT-rich interactive domain-containing protein 1B (*ARID1B*) (6q25) (30% versus 10%, FDR $= 0.058$), a member of the SWI/SNF-A chromatin-remodeling complex and cyclin-dependent kinase inhibitor 2C (*CDKN2C*) (1p32) (46% versus 24%, FDR $= 0.08$) genes were significantly more common in atypical *NF2* samples (Fig. 1e).

When we corrected both for the mutational background (*NF2* versus non-*NF2* mutant) and CNV status (CNV-high versus-low), we again did not identify any differences in the overall number of damaging somatic mutations, further suggesting that changes in coding variations did not significantly contribute to the risk of being atypical.

**mRNA expression signature separates atypicals from benigns**. After investigating genomic alterations, we expanded our analysis to understand the transcriptional changes underlying atypical meningiomas. We studied the messenger RNA (mRNA) expression profiles of 138 primary meningiomas, again focusing on the comparison between atypical versus benign samples ($n = 26$ versus 112, respectively) (Supplementary Data 4a). Principal component analysis, unsupervised hierarchical and consensus clustering clearly distinguished *NF2* CNV-low, *NF2* CNV-high and non-*NF2* samples, however failed to completely separate atypical versus benign tumours (Fig. 2a). This result suggested that gene expression correlated more closely with the underlying driver mutation rather than the histological grade.

To characterize specific transcriptional changes associated with atypical tumours, we next corrected for the underlying meningioma driver mutation by dividing the cohort into *NF2* and non-*NF2* subgroups as before. We identified mRNA signature genes ($n = 483$) that were differentially expressed between atypical and benign samples when stratified by *NF2* status (Supplementary Data 4b). Principal component (PC) analysis of

gene expression data using these signature genes correctly separated atypical samples from benign ones, even when *NF2* status was not considered (Fig. 2b; Supplementary Figs 6–9). This analysis demonstrated that atypical tumours harbour a distinct expression profile of a set of transcripts, which were significantly enriched for cell cycle processes, including upregulation of the E2F and FOXM1 transcription factor networks (hypergeometric test) (Fig. 2c,d; Supplementary Fig. 10; Supplementary Data 4c,d). Importantly, expression levels of the genes associated with atypical samples were not affected by the CNV events, suggesting that transcriptional regulation, and not genomic instability, was the primary driver of these processes.

We next used the top 25 most differentially expressed genes to build a random forest prediction model, aiming to associate gene expression with meningioma histological grade. Our prediction model had a 96% prediction accuracy (4% out of bag error rate) on this training set. When we used the dataset of an independent meningioma gene expression study[13] as a validation set, we obtained a 91% accuracy rate, after considering only histologically atypical meningiomas with high or medium Ki-67 index. Interestingly, 15 of the top 25 most significant genes were involved in cell cycle processes, including E2F, ASPM, aurora kinase, cyclin and centromic proteins (Fig. 2c; Supplementary Data 4b). Other top upregulated genes included *POLQ*, a DNA damage repair gene associated with high tumour grade and genomic instability in breast cancer[14]; *RET*, a receptor tyrosine kinase oncogene; and *BCL2*, which has been found to have increased expression in many types of cancer such as lymphoma, small cell lung and prostate cancer[15–17]. Importantly, atypical meningiomas also showed increased expression of the *EZH2* gene. This gene codes for the catalytic subunit of the PRC2 complex, known to play a key role in both tissue-specific stem cell maintenance and tumour development (Fig. 2d).

**microRNA regulatory networks in atypical meningiomas**. We next compared miRNA expression patterns in benign versus and atypical meningiomas ($n = 15$ versus 17, respectively) (Supplementary Data 5a). Unsupervised hierarchical clustering of miRNA expression profiles correlated well with being benign versus atypical, CNV status, and also with the underlying meningioma driver mutation, clearly separating *NF2* mutants from non-*NF2* meningiomas (Fig. 3a).

On the basis of this observation, we searched for differentially expressed miRNAs between atypical and benign meningiomas (Fig. 3b, Supplementary Data 5b). Differential expression of miRNAs has been shown to play important roles in the control of cancer hallmark functions such as invasion, metastasis, proliferation and apoptosis[18] (Fig. 3c). In atypical samples, we identified 67 differentially regulated miRNAs (54 downregulated, 13 upregulated). We next correlated the differentially expressed miRNAs with large-scale chromosomal events. We identified miRNAs clustering on a deleted region on chromosome 14q32 ($n = 28$) to be downregulated, suggesting the expression of these miRNAs were largely driven by CNVs.

To correlate downregulated miRNA clusters with gene expression patterns, we inferred miRNA:mRNA regulatory networks using samples that contained both datasets ($n = 22$). We calculated the observed number of negatively correlated targets for each miRNA, with the assumption that if the miRNA expression were playing a significant biological role, it is expected that the expression levels of its target genes would be altered. Consistent with the expected biological action of elevated miRNA expression, we identified the number of negatively correlated targets to be significantly higher than the positively correlated ones ($P < 2.2e - 16$, paired Wilcoxon test) (Supplementary

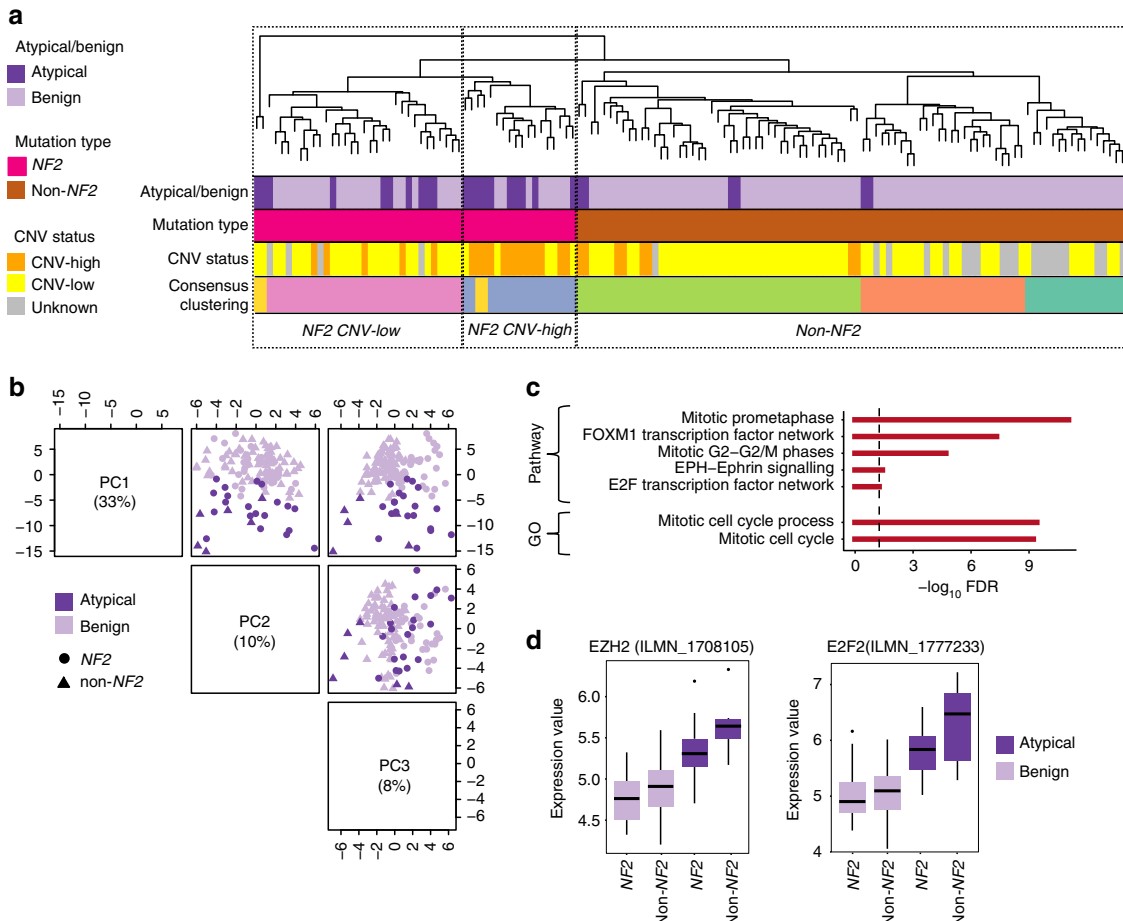

**Figure 2 | mRNA expression profile in atypical meningiomas. (a)** Unsupervised hierarchical clustering of 138 meningiomas by genome-wide expression profiling is shown. Atypical versus benign histology, underlying meningioma driver mutations, copy number variations, which are colour coded, are shown on the left. Although the expression profile accurately clusters meningioma samples based on driver mutations, it does not fully differentiate atypical versus benign tumours. Consensus clustering also separates non-*NF2* meningiomas (Hedgehog, *POLR2A*, *TRAF7/PI3K*, *TRAF7/KLF4*) into different subgroups. **(b)** PC analysis using mRNA signature genes separates atypical and benign samples. **(c)** GO Term and pathway enrichment for genes associated with the atypical phenotype compared with benign are shown (atypical $n = 26$, benign $n = 112$). Black line indicates an FDR of 0.05. **(d)** *EZH2* and *E2F2* gene expressions, which are upregulated in atypical versus benign samples, are plotted (empirical Bayesian method). Lines depict the median values; boxes plot 25th to 75th percentiles, whereas separately plotted dots show the outliers.

Fig. 11). We found that loss of miRNAs on the frequently deleted chromosomal segment 14q32 was significantly inversely correlated with a set of their known target genes ($n = 85$) (Supplementary Data 5c). These upregulated genes were enriched for receptor tyrosine kinase signalling pathways as well as the GO term 'regulation of programmed cell death' (Fig. 3d; Supplementary Data 5d).

We also identified 5 members of *let-7* family (*let-7a*, chr22, log FC = 0.66; *let-7b*, chr22, log FC = −0.8; *let-7c*, chr21, log FC = −0.6; *let-7d*, chr9, log FC = −0.78; *let-7e*, chr19, log FC = −0.86) to have aberrant expression in atypical meningiomas. This family of miRNAs has been shown to be deregulated in various cancers such as prostate cancer and neuroblastoma[19,20], as well as malignant schwannoma and meningioma[21], potentially through negative regulation of EZH2 (refs 19,20,22). Consistent with these observations, the down regulation of *let-7* family in atypical meningiomas was correlated with upregulation of *EZH2* mRNA expression in our dataset (*let-7c*: ∼ −0.65, $P = 0.001$, *let-7d*: ∼ −0.62, $P = 0.002$, *let-7e*: ∼ −0.49, $P = 0.02$ correlation test) (Fig. 3e). Importantly, expression levels of these miRNAs, other than *let-7b* (deleted in 75% versus 47% in atypical versus benign meningiomas, respectively) were not affected by the CNV events.

**DNA methylation patterns in atypical meningioma subgroups.** We next focused on epigenetic alterations, again comparing atypical meningiomas to benign ones. We initially analyzed the DNA methylation status of 60 samples, including meningiomas ($n = 46$ versus 11, benign versus atypical, respectively) as well as control tissues obtained from the normal meninges ($n = 3$) (Supplementary Data 6a,b). Unsupervised analysis of whole-genome methylation data using two different clustering methods, namely principal components and consensus clustering, revealed consistent results and identified distinct subgroups of meningiomas (Fig. 4a,b). Neither of these clusterings was affected by large-scale chromosomal events, since when we removed the sites in the regions that are affected by large-scale chromosomal events (chr1, chr14, chr22), clustering results did not change (Supplementary Fig. 12).

Integration of the methylation data with copy number and mutational analyses revealed the presence of distinct molecular signatures in atypical meningiomas. First, genome-wide DNA methylation patterns clearly separated the *NF2* mutant meningiomas from relatively less methylated non-*NF2* tumours, with atypical non-*NF2* meningiomas clustering distinctly from benign non-*NF2* ones (Fig. 4a,b). Second, for *NF2* mutant meningiomas, which also formed 2 different sub-clusters, one of

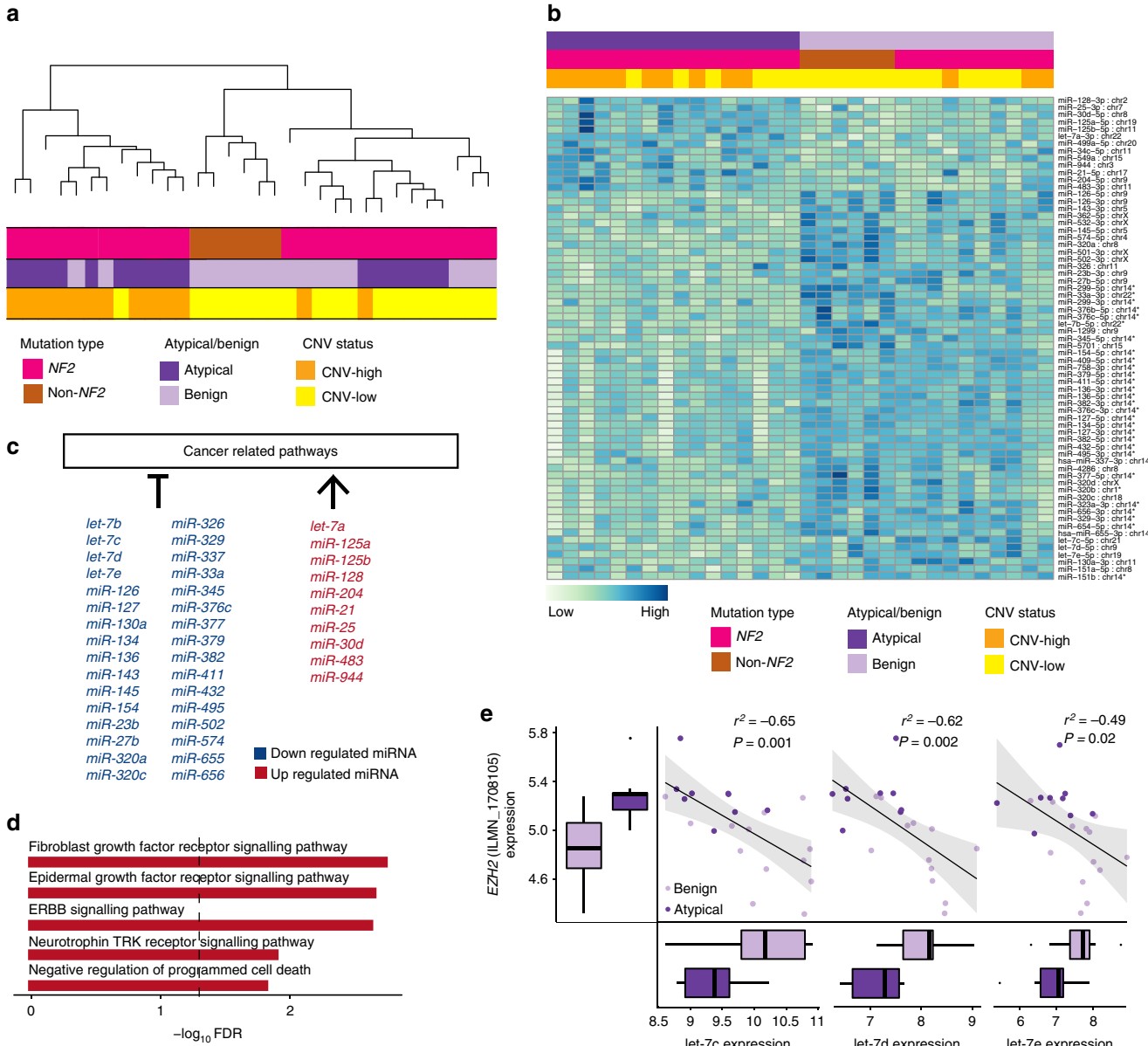

**Figure 3 | miRNA expression profile in atypical meningiomas.** (**a**) Unsupervised hierarchical clustering of genome-wide miRNA expression of 32 meningiomas. Integrated view of the miRNA expression clustering, combined with the underlying meningioma driver gene mutation, histological grade and CNV-high or -low status, which are all colour coded, are shown. (**b**) Heatmap visualization of differentially expressed miRNAs between atypical versus benign samples. The colour scale for $\log_2$ (fold change) is shown at the bottom. MiRNAs affected by CNVs are marked by an asterisk. (**c**) Differentially expressed miRNAs with either inhibitor or activator effects on cancer hallmark functions are shown. Red coloured miRNAs are upregulated whereas blue coloured miRNAs are downregulated in atypical samples. (**d**) The top gene ontology terms for the genes regulated by the 14q32 miRNA cluster. Black line indicates an FDR value of 0.05. (**e**) Expression of let-7 family, a known inhibitor of EZH2, negatively correlates with *EZH2* mRNA expression (Spearman correlation test). Lines depict the median values; boxes plot 25th to 75th percentiles, whereas separately plotted dots show the outliers.

the subgroups revealed a distinct hypermethylated phenotype and was significantly enriched for atypical CNV-high samples ($P = 0.03$ (two *NF2* cluster comparison), Fisher's exact test) (Fig. 4a–c; Supplementary Data 6b). The majority of samples in this cluster revealed large-scale chromosomal aberrations including chromosome 1p loss, which was found in all samples (Fig. 4a; Supplementary Data 6b). Indeed, we observed a statistically significant positive correlation between the degree of chromosomal alterations and the amount of genome-wide DNA hypermethylation ($P = 0.01$ correlation test) (Fig. 4d).

Interestingly, we found that atypical samples harbouring both *NF2* and *SMARCB1* mutations, although genomically stable, still clustered closer to the CNV-high atypical *NF2* samples in

principal component analysis and were also hypermethylated. These results suggest the presence of two distinct pathways underlying *NF2* mutant, atypical meningiomas: one through acquiring genomic instability and the other through recurrent *SMARCB1* mutations (Fig. 4b,c).

**Hypermethylation of PRC2 binding sites in atypical tumours.** Given that *NF2* mutant atypical meningiomas displayed a hypermethylated phenotype, we next investigated which gene sets were more methylated in atypical versus benign meningiomas (GREAT tool), making use of the Molecular Signature Database (MSigDB)[23]. We found that Polycomb Repressive Complex 2

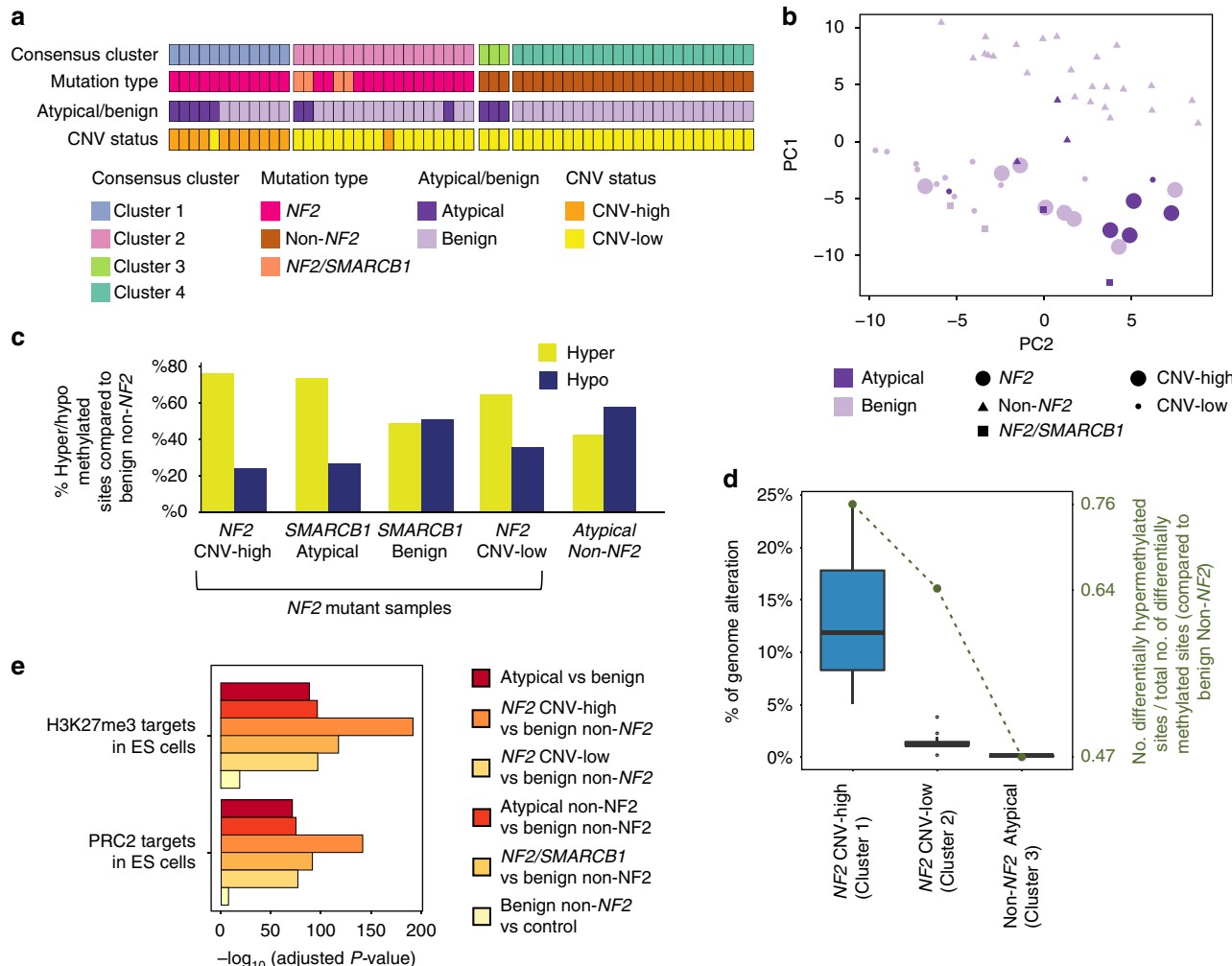

**Figure 4 | DNA-methylation patterns across benign versus atypical meningiomas. (a)** Integrated view of DNA methylation clustering combined with the underlying meningioma driver gene mutations, histological grades and genome copy numbers are shown (atypical $n = 11$, benign $n = 46$). Consensus clustering defines four main subgroups based on methylation status. The CNV-high or -low categories, as well as *NF2* and *SMARCB1* mutations are marked. **(b)** PC analysis of meningioma methylation data, which separate *NF2* CNV-high, -low and non-*NF2* samples into different groups, is shown (atypical $n = 11$, benign $n = 46$). **(c)** Percentages of hyper- or hypo-methylated sites among differentially methylated sites in various subgroups, as compared with benign non-*NF2* samples are plotted. All atypical as well as benign CNV-high *NF2* mutant meningiomas are hypermethylated when compared with benign non-*NF2* samples (atypical $n = 11$, benign $n = 46$). **(d)** Percentage of altered genome significantly correlates with the extent of genome-wide DNA hypermethylation (atypical $n = 11$, benign $n = 46$). Along the horizontal axis, the meningioma samples are grouped according to the consensus cluster numbers as in **a**. The number of differentially hypermethylated sites divided by the total number of sites is shown as the green dashed line and quantified along the vertical axis on the right. Lines depict the median values; boxes plot 25th to 75th percentiles, whereas separately plotted dots show the outliers. **(e)** Comparisons for atypical versus benign as well as various other molecular subgroups, which are colour coded, are shown for Molecular Signatures Database (MSigDB) gene set enrichment summary plots. The most significant gene sets are plotted.

(PRC2) target sites in human embryonic stem cells (PRC2-hESCs)[24], as well as Homeobox domain sites[25], represented the most differentially methylated regions (FDR = $1.15 \times 10^{-71}$ and $6.6 \times 10^{-55}$, respectively) (Fig. 4e; Supplementary Data 7a). Interestingly, we also found enriched methylation of these sites across benign meningiomas compared with control meninges, albeit less than that observed in atypical samples (FDR = $5.8 \times 10^{-17}$). This finding suggests that the degree of methylation at PRC2-hESCs and Homeobox domain sites may establish a spectrum of meningioma severity, with normal meninges, benign meningioma, and atypical meningioma having progressive amounts of increased DNA methylation at these sites.

To further explore this finding, we next considered the methylation status of PRC2-hESC targets among the various subgroups identified in our genomic, gene expression, and methylation analyses. We observed the highest enrichment for PRC2-hESCs hypermethylation in the atypical and benign *NF2* CNV-high samples (FDR $9.3 \times 10^{-142}$ when compared with benign non-*NF2* mutant samples), followed, in order, by the *NF2/SMARCB1*, *NF2* CNV-low and atypical non-*NF2* meningiomas (FDR values $1.2 \times 10^{-91}$, $4.1 \times 10^{-77}$ and $2.8 \times 10^{-75}$ respectively) (Fig. 4e; Supplementary Data 7b–f).

It has been previously shown that EZH2, the catalytic subunit of PRC2, is a recruitment platform for DNA methyltransferases (DNMTs), acting as a direct controller of DNA methylation at PRC2 binding sites[26]. On the basis of this knowledge and our observations, transcriptional upregulation of EZH2 and down-regulation of its regulator *let-7* in atypical samples suggest that increased methylation in atypical tumours might be related to

deregulated PRC2 activity. However, this association was not experimentally tested in our study. (Supplementary Figs 13 and 14; Supplementary Data 8a,b).

**H3K27me3 ChIP-seq confirms silencing of PRC2 targets**. In addition to DNA methylation, the PRC2 complex also plays an important role in regulation of histone state. In particular, EZH2 is involved in trimethylation of histone 3 lysine 27 (H3K27me3) to drive long-term silencing of gene expression during development and other processes[27]. To confirm increased activity of EZH2 in atypical tumours, we next performed H3K27me3 ChIP-seq using both atypical and benign meningioma samples ($n = 3$ each) (Supplementary Data 9a). H3K27me3 profiles clearly separated atypical meningiomas from benign ones (Fig. 5a). Interestingly, differential binding analysis between atypical and benign samples showed an overall increase in H3K27me3 binding in atypical samples (Fig. 5b; Supplementary Data 9a).

GO Term enrichment analysis of H3K27me3 differentially bound and underexpressed genes in atypical as compared with benign samples revealed 'neuron projection morphogenesis', 'neuron differentiation', and 'neurogenesis' GO terms, suggesting that atypical meningiomas, while turning on embryonically active pathways, repress pathways involved in differentiation (Fig. 5c, Supplementary Data 9b). On the basis of GREAT enrichment analysis, H3K27me3 differentially bound and underexpressed genes in atypicals as compared with benign ones were enriched for PRC2-hESC targets (FDR = 0.03). Since GREAT analysis assesses the enrichment using hypergeometric test, the enrichment was significant even though there was an overall increase in H3K27me3 signal in atypicals.

**H3K27ac ChIP-seq identifies activated regions in atypicals**. Given the role of epigenetic silencing in atypical samples, we investigated if activating epigenetic alterations may also be associated with these tumours. To investigate this question, we performed ChIP-seq targeting histone 3, lysine 27 acetylation (H3K27ac) using 3 atypical tumours and 15 benign tumours, as well as control meningeal tissues (2 dura samples) (Supplementary Data 10a). We identified transcriptionally active regions by focusing on broad genomic loci that harboured dense clustering of H3K27ac signals, previously described as 'super-enhancer' regions. These features have been shown to be important in determining not only cell identity, but also cancer cell properties[28]. Indeed, super-enhancer driven expression of particular oncogenes has been shown to be fundamental in formation of specific tumours[28]. Although active genomic regions as defined by H3K27ac binding were highly correlated among different meningioma samples (with a minimum correlation coefficient of 0.7), we were able to clearly classify meningiomas into atypical versus benign samples as well as into various molecular subgroups, including NF2 CNV-high, NF2 CNV-low and non-NF2 samples (Fig. 5d).

We identified 19 super-enhancers with concordant changes in gene expression between atypical and benign samples (Fig. 5e, Supplementary Data 10b). Notably, in the atypical group we identified a differentially active super-enhancer near the ZIC Family Member 1 (ZIC1) transcription factor, a regulator of neural crest differentiation[29], that is shown to play an essential role in the proliferation of meningeal cell progenitors[30]. This gene showed increased H3K27ac binding in atypical versus benign meningiomas (FDR = 0.007, fold change log FC = 4.95) with increased expression in atypical NF2 samples (gene expression FDR = 0.001) (Fig. 5f,g).

To characterize super-enhancers associated with atypical tumours independent of the mutational background, we next compared atypical NF2 mutant samples to benign NF2 mutants and identified five super-enhancers with concordant changes in gene expression. We found decreased H3K27ac signal near the GDNF family receptor alpha-1 (GFRA1) gene that is involved in neuron survival and differentiation[31]. Importantly, decreased H3K27ac binding was associated with decreased GFRA1 expression in atypical NF2 versus benign NF2 meningiomas (FDR $6.2 \times 10^{-5}$ fold change log FC = 9.54).

**Benign meningiomas undergoing atypical progression**. We next expanded our analysis of primary atypical meningioma to include recurrent (or progressed) atypical tumours. On the basis of our findings as well as previously published studies[11], we hypothesized that distinct molecular mechanisms may be involved in formation of de novo atypical meningioma versus progression of benign tumours. Our analysis focused on a small cohort of paired meningioma samples that included the original benign sample as well as the progressed atypical counterpart from the same patient ($n = 4$). On the basis of exome sequencing, we identified all samples to be NF2 mutant (Supplementary Data 11).

Consistent with the previously published literature[32], we found Telomerase Reverse Transcriptase (TERT) gene promoter mutations (C228T/-124G-A chr5:1,295,228; C250T/-146G-A chr5:1,295,250) in 2 of the 4 progressed atypical samples. A larger screen of 27 recurrent atypical meningiomas (non-paired) identified 2 additional samples, suggesting approximately 13% (4/31) of progressed atypical samples harbour mutations in the TERT promoter (Supplementary Data 11). Importantly, when we screened our cohort of 110 primary, non-recurrent meningiomas ($n = 66$ NF2 atypical, $n = 12$ NF2 benign CNV-high, $n = 4$ non-NF2 benign, CNV-high, $n = 12$ NF2 benign CNV-low, $n = 12$ non-NF2 benign CNV-low and $n = 4$ NF2/SMARCB1), we did not identify any TERT promoter mutations (Supplementary Data 11). This finding further supports the presence of two distinct molecular pathways that underlie formation of atypical meningiomas: either through a de novo pathway as detailed in this study, or due to transformation of benign meningiomas, partially through acquisition of an activating TERT promoter mutation.

## Discussion

Among the three pathological grades of meningiomas, grade II atypical tumours show increased mitotic activity as well as high cellularity, small cells with a high nuclear to cytoplasmic ratio, prominent nucleoli, uninterrupted patternless or sheet-like growth and occasional foci of spontaneous necrosis[2,3]. The 2007 revision of the WHO system classified meningiomas with brain invasion also as grade II, even if they were histologically benign, as the presence of brain invasion worsened the clinical outcome and often necessitating the need for adjuvant radiotherapy. Thus, the WHO grading system correlates well with clinical parameters including the likelihood and time to recurrence after initial treatment (that is, surgery) as well as overall survival, which worsens with increasing grade. Indeed, atypical meningiomas are associated with up to a 40% recurrence rate at 5 years following total resection[4]. Besides dictating the clinical course, pathological grading of meningiomas further guides treatment such that surgical resection is the standard first-line treatment for all accessible, symptomatic meningiomas, with radiation being reserved for the post-operative management of higher grade II and III tumours. There are no standard chemotherapy options.

Unlike grade I benign meningiomas, the genomic landscape of atypical meningiomas is not well understood. Here we studied the genomic landscape of histologically atypical, primary grade II

meningiomas only, excluding those tumours that progressed from a benign, recurrent meningioma or those that were classified as grade II based solely on brain invasion. Utilizing comprehensive next-generation genomic approaches, including exome sequencing, mRNA and miRNA expression profiling, as well as H3K27 acetylation and trimethylation ChIP-seq and DNA methylation analyses, we identified the molecular features associated with

atypical samples independent of their mutational background and CNVs. We found that primary atypical meningiomas were enriched for *NF2* loss. On the basis of exome sequencing, we did not observe any significant difference in the number of somatic protein altering mutations in atypical versus benign samples (even when stratified by *NF2* status), and also did not find novel recurrent driver genes.

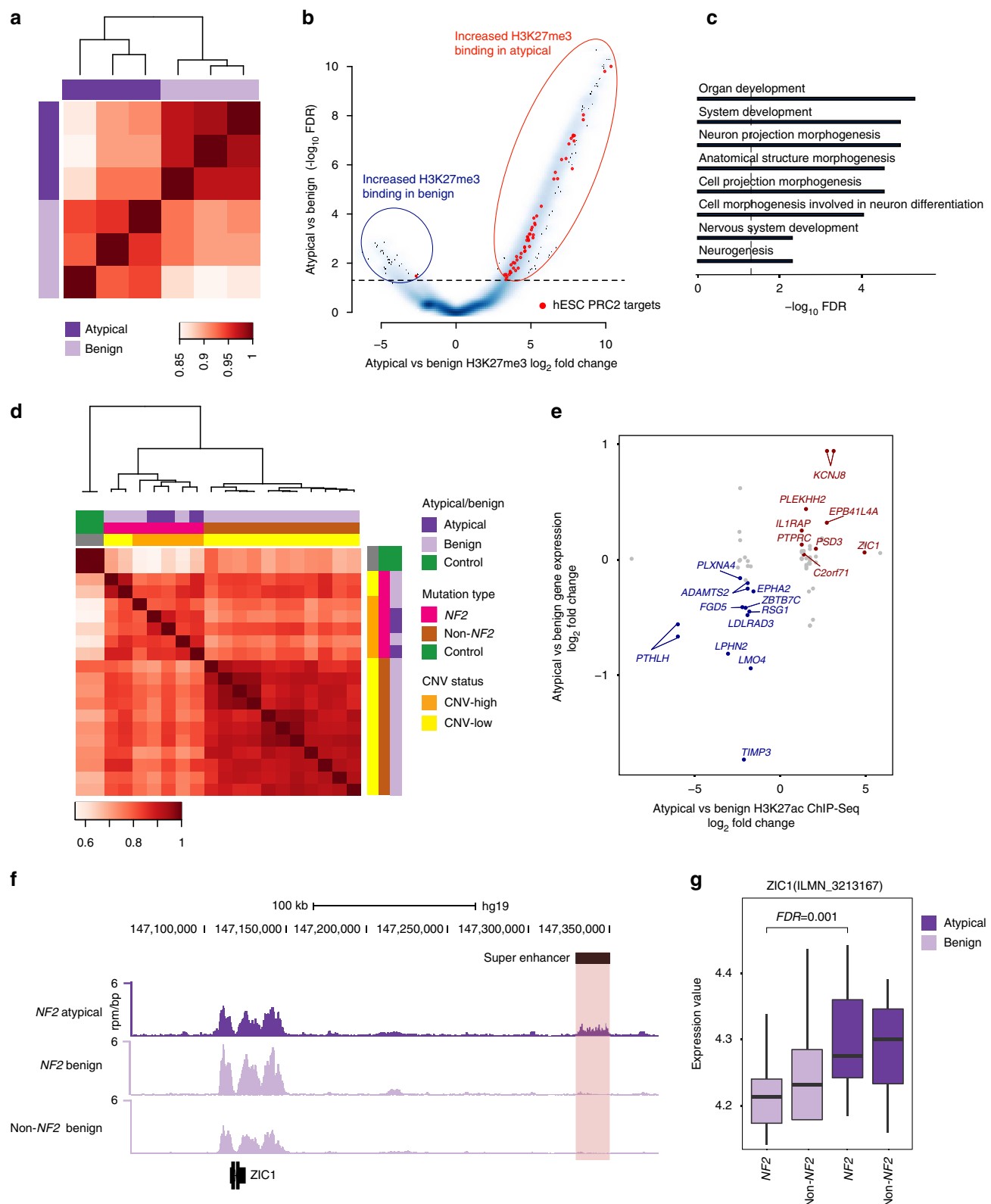

However, we did find significant differences in the extent of chromosomal instability, with genomically unstable samples having the greatest chance of being *NF2* mutant atypical. The CNV status associated significantly with levels of DNA methylation, such that CNV-high *NF2* mutant meningiomas displayed a hypermethylated phenotype, particularly affecting the PRC2 targets in embryonic stem cells. These sites also revealed increased H3K27me3 signals based on ChIP-seq. These observations raise the possibility that differentiation-related genes may be suppressed in atypical meningiomas, similar to PRC2-mediated inhibition of these genes during embryonic development[33]. Indeed, silencing of PRC2 targets has been shown to play a role in the formation and maintenance of other forms of cancer by locking the cells into stem-like cellular state[11,34]. We also identified the Homeobox genes, which similar to PRC2 signalling play a role in embryogenesis and differentiation[35], to be hypermethylated in *NF2* CNV-high meningiomas.

Consistent with these findings, we showed transcriptional upregulation of the catalytic subgroup of the PRC2 complex, EZH2, in atypical meningiomas as compared with benign ones. Besides functioning as a histone methyltransferase, EZH2 has been shown to be a recruitment platform for DNA methyl-transferases (DNMTs), acting as a direct controller of DNA methylation at PRC2 binding sites[36]. Previous reports have also identified this enzyme as a marker for aggressiveness, particularly in glioblastoma and renal cell carcinoma[37,38]. The association of EZH2 expression with atypical meningiomas suggests a role for EZH2 as a marker in higher grade meningiomas.

A potential mechanism for increased EZH2 expression is the loss of the miRNA *let-7*, a known negative regulator of EZH2, in atypical samples (*let-7c*: FDR = 0.03, *let-7d*: FDR = 0.01, *let-7e*: FDR = 0.01). Indeed, samples with *let-7* loss revealed increased EZH2 mRNA expression (*let-7c*: $\sim -0.65$, $P = 0.001$, *let-7d*: $\sim -0.62$, $P = 0.002$, *let-7e*: $\sim -0.49$, $P = 0.02$, correlation test) in atypical compared with benign samples. This finding suggests that loss of *let-7* contributes to increased EZH2 expression and deregulated PRC2 activity in atypical meningiomas. We also identified miRNA cluster at 14q32 to be downregulated in atypical samples, affecting genes in receptor tyrosine kinase signalling pathways. Deregulation of miRNA cluster at 14q32 has been reported in various cancer types[39].

In a distinct subgroup of CNV-low but hypermethylated *NF2* mutant atypical meningiomas, we observed recurrent *SMARCB1* mutations. Interestingly, loss of the tumour suppressor *SMARCB1* has previously been reported in various malignant rhabdoid tumours[40], as well as in families with multiple meningiomas with schwannomatosis[41]. Our group has also recently described recurrent *SMARCB1* mutations in a subset of benign

meningioma[7], however enrichment of this alteration in a distinct molecular subgroup of atypical samples has not previously been reported.

Taken together, we report a unique set of genomic and epigenomic events that distinguish primary atypical tumours from other types of meningiomas. Similar to the molecular pathways underlying formation of malignant gliomas, we propose that distinct molecular mechanisms underlie the formation of *de novo* (primary) and progressed (recurrent) atypical meningiomas. While previous groups have reported *TERT* promoter mutations in atypical samples, we find that these

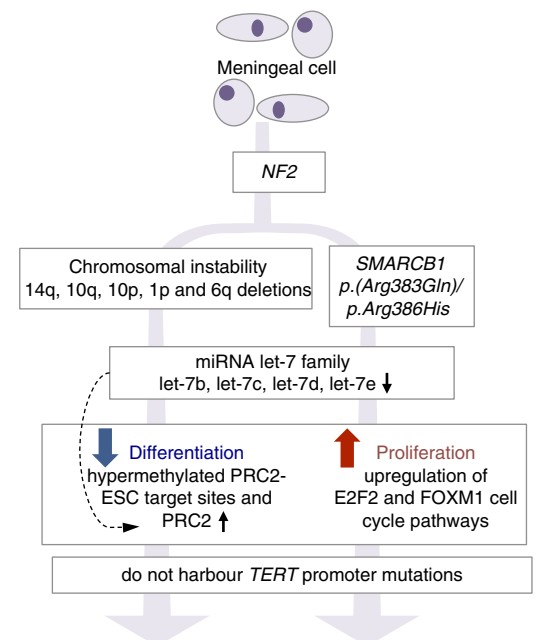

**Figure 6 | *De novo* pathway in the formation of atypical meningiomas.** Samples with *NF2* loss were highly enriched among primary (*de novo*) atypical cases, which co-occur either with genomic instability or recurrent mutations in *SMARCB1*. These tumours have hypermethylation of the PRC2 binding sites in hESCs, thereby phenocopying a more primitive cellular state. Epigenetic changes acting on the PRC2 networks are associated with upregulation of the catalytic subunit EZH2, and down regulation of its regulator *let-7*. Primary atypical tumours harbour increased expression of cell-cycle related genes, including the E2F and FOXM1 transcription factor networks. TERT promoter mutations are not present in *de novo* tumours, and limited to recurrent samples only.

**Figure 5 | Histone modifications in meningiomas.** (**a**) Unsupervised hierarchical clustering of meningioma samples using H3K27me3 profiles is shown (Pearson correlation) (atypical $n = 3$, benign $n = 3$). Grades are colour coded. (**b**) Starburst plot for atypical versus benign H3K27me3 ChIP-seq signal fold change (horizontal axis) and FDR (vertical axis) is shown. Red and blue circles indicate regions that have increased binding in atypical versus benign meningiomas, respectively. PRC2-hESC targets are marked in red colour. These analyses reveal increased H3K27me3 binding in atypical samples, which shows particular enrichment of PRC2-hESC targets. (**c**) GO Term enrichment plot for genes with increased H3K27me3 binding and decreased expression is shown (atypical $n = 3$, benign $n = 3$). Black line indicates an FDR of 0.05 (**d**) Unsupervised hierarchical clustering of 18 meningiomas and 2 controls based on H3K27ac profile in super-enhancer sites is shown (Pearson correlation). Super-enhancer profiles are highly correlated among different meningioma subtypes. All samples are colour coded, which are shown on the right. (**e**) H3K27ac ChIP-seq results correlate with gene expression profiles ($n = 18$). Atypical versus benign H3K27ac ChIP-seq signal fold change is plotted along the horizontal axis, whereas atypical versus benign gene expression fold change is shown along the vertical axis. Red dots indicate genes that have both activated super-enhancers and are overexpressed in atypical versus benign meningiomas, whereas blue points indicate genes that are associated with de-activated enhancers and are underexpressed. An FDR threshold of 0.05 is used both for gene expression and H3K27ac ChIP-seq data (empirical bayes and DiffBind methods). (**f**) H3K27ac ChIP-seq occupancy at a super-enhancer near *ZIC1* ($n = 18$). The horizontal axis shows genomic position whereas the vertical axis shows signal of ChIP-seq occupancy in units of reads per million (r.p.m.). Super-enhancer region that is differentially bound in atypical samples is depicted as a black line over the gene track and highlighted. (**g**) *ZIC1* gene expression across atypical and benign samples is plotted ($n = 138$) (empirical bayes method). Lines depict the median values; boxes plot 25th to 75th percentiles, whereas separately plotted dots show the outliers.

events were limited to recurrent samples only, and not present in *de novo* tumours. Our results show that primary atypical lesions arise in response to synergistic signals that stimulate proliferation (through upregulation of the E2F2 and FOXM1 regulated cell-cycle pathways), as well as epigenetic changes (through EZH2/PRC2 target activation), that have previously been shown to phenocopy more primitive cellular states[34] (Fig. 6).

In conclusion, we report comprehensive genomic, transcriptomic and epigenomic analyses of primary atypical meningiomas, identifying distinct molecular pathways that define the landscape of these tumours and separate them from recurrent atypical samples. Our results have biological as well as clinical implications through enhanced molecular classification and therapeutic intervention in atypical meningiomas.

## Methods

**Clinical materials.** Institutional Review Board approvals for genetic studies, along with written consent from all study subjects, were obtained at the participating institutions. The specific approval committees included the Human Investigation Committee at Yale University, Ethikkommission der Medizinischen Fakultät der Universität zu Köln and Ethikkommission an der Medizinischen Fakultät der Rheinischen Friedrich-Wilhelms-Universität Bonn.

**Selective tissue dissection.** For each frozen specimen submitted for whole-exome sequencing, sections were re-reviewed to confirm the diagnosis and assess the adequacy of the frozen tissue for experimental analysis. On H&E-stained sections from frozen tissue blocks, areas of interest were identified and microscopically dissected to ensure that each sample consisted of >70% tumour cells; unwanted regions such as inflammatory and necrotic areas were excluded. Tumours in the replication cohorts did not undergo selective tissue dissection. DNA/RNA was prepared using the Allprep DNA/RNA Mini Kit (Qiagen) with the assistance of a QIAcube.

**Exome capture and sequencing.** Nimblegen/Roche human solution-capture exome array (Roche Nimblegen, Inc.) was used to capture the exomes of blood and tumour samples according to the manufacturer's protocol. Sequencing of the library was performed on Illumina HiSeq instruments using paired-end 74 basepair reads by multiplexing two tumour samples or three blood samples per lane. Image analysis and base calling was performed by Illumina Pipeline with default parameters, installed on Yale University's High Performance Computing Cluster.

**Whole-exome sequence analysis.** We first filtered the reads based on Illumina quality score. The low-quality 3′-end of the reads using FASTX-Toolkit (http://hannonlab.cshl.edu/fastx_toolkit/index.html) and PCR primer-contaminated sequences, that are considered to lead to alignment artifacts, were trimmed using cutadapt. The sequences were kept only if both reads in a pair had more than 35 bases remaining after the above trimming and filtering quality reads. The reads were aligned to the human reference genome (version GRCh37) using Stampy (version 1.0.21) in a hybrid mode with BWA (version 0.5.9-r16)[42,43]. MarkDuplicates algorithm from Picard was used for flagging PCR duplicates. Alignment quality metrics were calculated using CollectAlignmentSummaryMetrics and CalculateHsMetrics utilities of Picard (http://picard.sourceforge.net/). Mean target coverage was 246 and 160 for tumour and blood respectively. The average percentage of reads with at least 20× coverage was 93% for tumour and 90% for blood respectively. We performed multi-sequence local realignment around putative and known insertion/deletion sites. This was followed by the base quality score recalibration using the Genome Analysis Toolkit (GATK, version 2.5–2)[44]. We detected variant sites (point mutations and small indels) for tumour and matched blood pairs using the HaplotypeCaller algorithm from GATK. The genotype likelihood-based somatic score, which was proposed by Li, was used[45]. We filtered out the variants according to the classes of genotype changes in tumour with respect to the blood[46]. We used various quality metrics to filter out variants: (1) somatic score less than 20, (2) overlapping a RepeatMasker or segmental duplication annotated region, (3) low quality (<30) and low quality-by-depth values (<1) (4) high mapping quality zero reads, (5) strand bias, (6) mutation cluster of size >2 (7) homopolymer runs of length >=10 base pairs within +/− 5 base pairs around the mutation or from the right of the mutation or (8) ClippingRankSum (calculated by GATK) < −3 or >3. We also excluded the sites that have more than 1% frequency in the NHLBI Exome Variant Server Database (http://evs.gs.washington.edu/EVS/) and 1000 Genome Database. In addition we used our internal database of 2216 exomes to compare the variant allele frequencies of each gene and excluded the variants in the genes that have greater than 150 variant alleles. Finally, we annotated variant alleles using Ensembl database (version 69) with the help of Variant Effect Predictor (v2.7) tool[47]. From these functional annotations, we selected the most-deleterious consequence

for each variant site and considered the variant allele to be deleterious if its consequence was annotated as transcript ablation, splice acceptor/donor, stop gained, frame shift, stop lost and splice region, initiator codon, nonsynonymous codon predicted to be deleterious/damaging, or in frame codon loss/gain. We also called somatic point mutations and indels with MuTect[48]/Indelocator (http://archive.broadinstitute.org/cancer/cga/indelocator) and Strelka[49]. Since MuTect can only detect somatic point mutations it is used together with short indel variant calling method Indelocator. The co-occurrence and mutual exclusivity of genes were assessed using one-sided Fisher's exact test. The MutSig algorithm was used for determining the genes that were mutated more often than expected by chance with FDR <5% (ref. 50). Mutational signatures of 6 main categories, $T > C/A > G$, $A > C/T > G$, $G > C/C > G$, $C > T/G > A$, $A > T/T > A$, $G > T/C > A$) are calculated based on filtered somatic variants.

**Clonality analysis.** Clonality rate of each somatic mutation was calculated based on the variant allele frequency and ploidy at that site, taking into account the admixture rate of each tumour.

**CNV identification from exome data.** The log ratio of depth of coverage between tumour and blood was calculated using GATK-*Depth Of Coverage* tool. CNV segments were then called from the log ratio of depth of coverage using ExomeCNV R package[51]. False positive CNV events were corrected by calculating minor allele frequencies (BAF) in each CNV segment. In each CNV segment, B-allele frequencies (BAF) at heterozygous sites should deviate from 0.5 by at least 0.05 units. We estimated the admixture rate based on CNV analysis of paired tumour and blood samples. Copy number loss regions were extracted and for those regions the BAF of each tumour SNP that was heterozygous in blood was calculated. Finally the admixture rate was estimated from the degree of deviation from homozygosity using the qpure R package[52].

**Custom molecular inversion probe sequencing and analysis.** Targeted sequencing of exomic regions and exon-intron boundaries of *NF2*, *SMARCB1*, *TRAF7*, *PIK3CA*, *PIK3R1*, *PRKAR1A*, *SMO* and *SUFU* plus the recurrent variants AKT1 p.Glu17Lys and KLF4 p.Lys409Gln was performed using molecular inversion probes (MIPs). Recurrent mutations in *POLR2A* were assessed with Sanger screening.

Custom amplicon sequencing and analysis. Libraries consisting of the coding exons from *TRAF7*, *NF2*, *SMO*, and the recurrent mutations for AKT1 p.Glu17Lys and KLF4 p.Lys409Gln were created using the TargetRich custom amplicon kit (Kailos Genetics).

**Sanger sequencing.** Coding variants detected by whole-exome sequencing or targeted next-generation sequencing were confirmed by Sanger sequencing using standard protocols.

**Whole-genome genotyping.** The Illumina Platform was used for WGG and CNV analyses of the samples. Human OmniExpress-12v1.0 BeadChips that contain 733,202 markers were used according to the manufacturer's protocol (Illumina, San Diego, CA, USA). CNVs were detected by comparing the normalized signal intensity between tumour and matched blood or tumour and the average of all blood samples. Segmentation was performed on log intensity (R) ratios using DNACopy algorithm[53]. Large-scale chromosomal deletion or amplification was defined as affecting more than one-third of the chromosomal arm, whereas focal event deletion or amplification was defined as affecting less than one-third and more than one-tenth of the chromosomal arm with accompanying log ratio of signal intensities < −0.1 or >0.1 and B-allele frequencies (BAF) at heterozygous sites deviating from 0.5 by at least 0.05 units. Large-scale copy neutral LOH was defined similarly, with the exception of log ratio of signal intensities being between 0.1 and 0.1. Percentage of genome alteration (PGA) was defined as the percentage of loss or gain base pairs relative to the entire genome. To identify significantly altered genes we applied Wilcoxon test between atypical and benign tumours using intensity log ratios. Cancer genes were identified based on the COSMIC database[54].

**Gene expression data.** We used Illumina HumanHT12.v4 chips on the gene expression data. Data was normalized using normal-exponential convolution model-based background correction and quantile normalization using the limma R package[55]. All batches were normalized at once after excluding probes with low quality. Samples estimated to have zero proportion of expressed probes, mean signal intensity being less than 5.54, or RIN value <3 were excluded. Hierarchical clustering of the gene expression data showed batch effect in the data. The batch effect was removed using ComBat in sva R package[56]. We performed unsupervised hierarchical clustering based on a Euclidean distance metric and average linking clustering on the probes that showed the top 1,000 most variable gene expression levels. GO term enrichment analysis was performed using the Cytoscape Reactome Plugin. Differentially expressed genes were identified using an empirical Bayesian method ebayes implemented in limma R package[55]. Genes were considered differentially expressed with adjusted *P*-value < 0.05.

**Random forest prediction model using gene expression data.** We used the randomForest R package (https://cran.r-project.org/web/packages/randomForest/index.html) for building a model to predict the histological grade[57]. We first performed feature selection by identifying the top 25 genes associated with histological grade from differential gene expression analysis. We trained the model on our top 25 differentially expressed genes using randomForest function with ntree = 5,000 parameter. Out of bag error rate (OOB), which is the error rate of the trained algorithm on a left out dataset that is not used in trained algorithm, was calculated. We next validated our prediction model on an independent dataset[13]. Raw Affymetrix data of the validation set was processed using the RMA method in affy R package (https://www.bioconductor.org/packages/devel/bioc/manuals/affy/man/affy.pdf).

**miRNA sequencing analyses.** Sequencing was performed on Illumina HiSeq instruments using 75 basepair, single-end reads and multiplexing 8 tumour samples per lane. Adaptor sequences were trimmed using cutadapt[58]. Reads shorter than 18 basepair and reads with quality less than 20 were filtered. Per base quality scores were assessed using FASTX tool. Adaptor trimmed reads were aligned to miRBase using mirDeep2 tool[59].

The set of known human miRNA precursors were downloaded from miRBase version 20 (ref. 60). We reported total read counts for 5p and 3p strands. Read counts for each sample were normalized to reads per million reads (RPM) and log$_2$-transformed. Batch effect was corrected using ComBat in sva R package[56]. We then performed unsupervised hierarchical clustering based on a Euclidean distance metric and complete linking. Differentially expressed miRNAs were identified using limma R package. An FDR threshold of 0.05 was used.

Six algorithms were used for miRNA target prediction: Miranda, Mirbase, Mirtarget2, Pictar, Tarbase and TargetScan using the RmiR.Hs.miRNA R package[61]. Regulatory targets of individual miRNAs were defined as those genes having significant negative correlation with the miRNA (spearman correlation test $P < 0.05$) and prediction support in at least three databases. GO term enrichment analysis was performed using the GOStats R package[62].

**DNA methylation data.** We performed DNA methylation profiling on 57 tumour samples and 3 control samples using the Illumina Infinium HumanMethylation450 BeadChip, which assesses the level of methylation at over 450,000 CpG sites across the entire genome (covering 99% of RefSeq genes and 96% of CpG islands). We first processed raw intensity files (*.idat) and obtained ratio between Illumina methylated probe intensity and total probe intensities (beta-values) using the champ package[63]. Then, we assessed the quality of the methylation samples and probes. Finally batch effect is corrected using combat algorithm in champ R package[63].

**Quality control and preprocessing for DNA methylation data.** We removed sites containing missing values. The probes targeting a CpG with a SNP were also removed from analysis. Probes targeting the X and Y chromosomes were excluded. The sites having at least 50% samples with detection $P$-value $> 0.05$ are removed. The detection $P$-value is calculated using the reported background signal levels of both the methylated and unmethylated channels. After preprocessing the raw data, we performed beta mixture quantile normalization (BMIQ), a normalization correction for the technical differences between the Type I and Type II array designs[23]. The DNA methylation score for each probe is described as the ratio of intensities between methylated and unmethylated alleles: M/(M + U).

**Unsupervised clustering for DNA methylation.** We performed consensus clustering on methylation $b$-values with 80% subsampling over 1,000 iterations of hierarchical clustering based on a Pearson correlation distance metric and average linking[64].

**Identifying differentially methylated sites.** Differentially methylated sites were calculated using the empirical bayes method called limma[55]. Sites were considered statistically significantly different between the groups if they had an adjusted $P$-value $< 0.05$ and median $b$-value difference $> 0.1$ or $< (-0.1)$. We used the GREAT tool, which internally maps genomic regions to genes and statistically controls for the fact that genes differ in size and their relative distance to each other[23]. Raw reads of EZH2 H1hesc ChIP-seq were downloaded from UCSC Genome Browser ENCODE database[65]. EZH2, ChIP-seq peaks were downloaded from Cistrome website[66].

**Immunofluorescence staining.** Meningioma frozen tissue sections were washed with phosphate buffer saline (PBS) for 5 min, then placed in 4% formaldehyde in PBS for 3 min (fixation), rinsed in PBS and permeabilized with 0.3% Triton-X100 in PBS for 40 s. The sections were rinsed with PBS 3 times for 5 min each and were incubated in BSDSGS blocking solution (PBS with 1% bovine serum albümin, 5% donkey serum, 5% goat serum, 0.1% glycine, 0.1% lysine) with 0.1% Tween 20. Afterwards, the sections were stained with 1:100 rabbit polyclonal anti-EZH2 antibody (5246P, Cell Signaling, Danvers, MA, USA). After washing, the slides were

incubated 30 min with 1:200 donkey-anti rabbit Alexa Fluor 555 (A315772, Life Technologies, Grand Island, NY, USA). After washing, slides were mounted with Vectashield DAPI medium (H1200, Vector Labs, Burlingame, CA, USA). Images were analyzed under an inverted microscope (Axio Vert A1, Zeiss, Obekochen, Germany) with fluorescent light source (X-Cite 120Q, Lumen Dynamics).

**H3K27ac and H3K27me3 ChIP-seq.** Briefly, 10 to 15 frozen sections of each tumour block or dura sample were collected for ChIP-Seq experiments. Tissue was crosslinked with 1% formaldehyde, quenched with glycine, and washed with PBS. Nuclei were extracted by dounce homogenization and resuspended in nuclear lysis buffer containing 0.3% SDS. Chromatin was sheared by sonication with a Q800R2 Sonicator by QSonica (60 min total, amplitude 30, 10 s pulses, 10 s rest). Soluble chromatin was incubated with magnetic beads coated with either H3K27ac antibody (ab4729) or H3K27me3 antibody (ab6002) overnight at 4 °C. Chromatin was precipitated using a magnet, washed extensively, and eluted with TE + 1% SDS. Crosslinks were reversed, purified, and subjected to standard Illumina paired-end multiplexed library construction. H3K27ac and H3K27me3 ChIP and input samples were sequenced for each tumour (1 × 75 bp, HiSeq 2000). H3K27ac and H3K27me3 reads were aligned uniquely with bowtie (0.12.7)[67] to the human genome (hg19). In H3K27ac, regions of enrichment were identified with MACS (v1.4)[68] whereas in H3K27me3, regions of enrichment were identified with MUSIC[69]. We used DiffBind to estimate significance of super-enhancer read change (adjusted $P$-value threshold = 0.05) between meningioma subtypes[70].

**Identifying super-enhancers from ChIP-seq data.** To identify super-enhancers, first regions enriched in ChIP-seq reads for H3K27ac were identified using MACS with input control. Super-enhancers were separated from typical enhancers using ROSE pipeline (https://bitbucket.org/young_computation/rose) with parameters -s (stitching) 12,500, -t (promoter exclusion zone) 2000. We used DiffBind to estimate significance of super-enhancer read change (adjusted $P$-value threshold = 0.05) between meningioma subtypes[70]. Cancer genes are defined based on COSMIC databases[54].

**Data availability.** All somatic mutations identified through exome sequencing of meningiomas were submitted to the COSMIC database previously[7]. Gene expression data is deposited in GEO database (accession: GSE84263)[7]. Our novel datasets including DNA methylation, miRNA sequencing, H3K27ac and H3K27me3 ChIP-seq are deposited in GEO database (accession: GSE91376).

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

## Acknowledgements

This study was supported by Gregory M. Kiez and Mehmet Kutman Foundation and Yale School of Medicine funds. Partial funding was provided through a research agreement between Gilead Sciences, Inc and Yale University. V.E.C. and M.W.Y. are supported by NIH Medical Scientist Training Program Grant T32GM007205. We gratefully acknowledge Arif O. Harmanci for constructive comments and discussions on the study.

## Author contributions

A.S.H. performed exome CNA, SNP array, DNA methylation array, ChIP-seq and miRNA expression analysis. A.S.H., E.Z.E.-O. and K.Y. performed mRNA expression analysis. A.S.H., E.Z.E.-O., V.E.C. and K.Y. performed whole-exome sequencing analysis. V.E.C., D.D.M., L.K. and M.W.Y. performed targeted next-generation sequencing, Sanger confirmation and mutation profile/clinical characterization. E.Z.E.-O. performed clonality analysis. V.E.C., O.H., G.C.-G., M.W.Y. aided in the selection of and prepared samples. A.S.H. and B.J.A. performed super-enhancer analysis. M.S., J.S. and B.K. provided samples and clinical data. M.W.Y., S.C. and V.E.C. performed ChIP-Seq experiments. K.B. supervised genomic experiments. O.H. and B.B. conducted immunofluorescence staining experiment. A.S.H. and M.G. wrote the manuscript. M.G. designed and oversaw the project.

## Additional information

**Competing financial interests:** Partial funding for sequencing of the tumour samples and partial salary support for S.C. and K.M.-G. was provided through a research agreement between Gilead Sciences, Inc. and Yale University. All other authors declare no competing financial interests.

**DOI: 10.1038/ncomms16215**      **OPEN**

# Author Correction: Integrated genomic analyses of *de novo* pathways underlying atypical meningiomas

Akdes Serin Harmancı, Mark W. Youngblood, Victoria E. Clark, Süleyman Coşkun, Octavian Henegariu, Daniel Duran, E. Zeynep Erson-Omay, Leon D. Kaulen, Tong Ihn Lee, Brian J. Abraham, Matthias Simon, Boris Krischek, Marco Timmer, Roland Goldbrunner, S. Bülent Omay, Jacob Baranoski, Burçin Baran, Geneive Carrión-Grant, Hanwen Bai, Ketu Mishra-Gorur, Johannes Schramm, Jennifer Moliterno, Alexander O. Vortmeyer, Kaya Bilgüvar, Katsuhito Yasuno, Richard A. Young & Murat Günel

*Nature Communications* 8:14433 doi: 10.1038/ncomms14433 (2017) Published online 14 Feb 2017; Updated 20 Apr 2018

In this Article, a subset of the H3K27ac ChIP-seq data (15 benign meningiomas and 2 dura samples (Sample IDs: MN-297, MN-288, MN-292, MN-163, MN-1037, MN-105, MN-201, MN-249, MN-191, MN-1066, MN-169, MN-291, MN-24, MN-79, MN-1044, CONTROL1, CONTROL2) was reported previously in a publication by the corresponding author[1]. These data were created by Dr. Justin Cotney in Dr. James Noonan's laboratory at Yale. The GEO database entry associated with this dataset has been updated to reflect this fact (GSE91372).

1. Clark *et al.*, Genomic analysis of non-NF2 meningiomas reveals mutations in TRAF7, KLF4, AKT1, and SMO. *Science* **1,** 1077–1080 (2013).

