## [Peer Review File · Nature Communications]

Reviewers' comments:

Reviewer #1 (Remarks to the Author): Expert in meningiomas

Manuscript NCOMMS-16-18001 "Integrated genomic analyses of de novo pathways..." by Harmanci et al.

The aim of the current study was to dissect molecular features defining primary de novo atypical (grade II) meningiomas. They performed next generation or targeted sequencing of 775 meningiomas, including 468 primary grade I and 88 primary atypical meningiomas. 75% of the grade II tumors contained NF2 mutations and 9% TRAF7/PI3K mutations. Targeted sequencing revealed a fraction of NF2 -mutated atypical meningiomas with simultaneous SMARCB1 mutations. They conclude that NF2/SMARCB1 mutation defines atypical phenotype. Moreover, primary NF2-mutated grade II meningiomas were characterized by high copy number variation (CNV). NF2 mutant tumors have greater chromosomal instability. Another feature of atypical tumors was chromosome 14q loss. Specific genes associated with chromosomal losses were PTEN, NOTCH2, MAX, and few others.

Analyzing the mRNA profile the authors found that atypical tumors have upregulation of E2F and FOXM1 transcription factor networks.

NF2-mutated and non-mutated meningiomas had a different miRNA profile, with let 7 family members being prominent.

Analysis of the DNA methylation status from 60 samples revealed atypical meningioma subgroups.

Another interesting finding was PRC2 hypermethylation in atypical and benign high-CNV meningiomas.

They also studied a subgroup of progressive meningiomas. Paired samples from 4 patients (all NF2 mutant).

In primary, non-recurrent meningiomas, no TERT promoter mutation was detected. The authors propose two different pathways for the development of atypical meningiomas - either de novo or due to transformation of primary benign meningiomas through TERT promoter mutations.

Comments:

The authors performed a comprehensive study using an impressive number of samples. However, a fraction of their observations has been already described before, including the genetic instability of NF2-mutated tumors, the NF2-SMARCB1 association, the chromosomal losses, and some expression data. Thus, some results have more confirmatory character than being new.

Some questions remain:

1. We know from clinical experience that atypical meningiomas can be very different in clinical course. So it would be highly interesting (and would add clinical relevance to the manuscript) to relate some features (CNV, others) to the time to tumor recurrence or the presence/absence of atypical meningioma recurrence. We see a fraction of atypical meningiomas which have no aggressive biology at all.

2. Are there any NF2 patients in the group?

3. At least a few cases with multiple (atypical) meningiomas should be studied

4. Atypical meningiomas are usually treated by irradiation after surgery. It would be interesting to compare non-irradiated vs. irradiated atypical tumor to see if a certain molecular feature predicts

the need for irradiation.

5. The histology of the "benign" tumors should be taken into account – how close are the fibrous grade I with NF2 mutation to atypical tumors (with/without NF2 mutation)?

Minor comments

-Ref.2: The 2007 WHO classification is no longer valid because a new classification has been published recently

-what is known regarding the let-7 miRNA family in meningiomas?

-supplementary figure 2 has some strange characters above the figure

Reviewer #2 (Remarks to the Author): Expert in brain cancer genomics

The authors carry out genomic/epigenomic analyses on subsets of a large set of meningiomas. They find that atypical (malignant) tumors are characterized by NF2 mutations, genomic instability, CpG hypermethylation, and increased K27me3 levels.

The above results are interesting and appear solid. However, many of the analyses presented in the manuscript do not provide additional support to the role of epigenetic reprogramming and many of the presented results are simple necessary outcomes of well-known relationships between chromatin states and transcription. The manuscript needs to clearly distinguish between primary findings (as enumerated in the paragraph above) and descriptive analyses, such as e.g. finding downregulated genes to be associated with increased K27me3, which is well known, and will be true in any comparison.

Most of the abstract and discussion are OK, but a substantial part of the result section needs to be toned down/removed to clearly indicate which analyses are as consistent with general known relationships between chromatin and expression, as they are with involvement of chromatin modifications in meningioma.

1. Abstract "Based on comprehensive genomics, transcriptomics and epigenomic analyses of 775 meningiomas" – This is misleading, please rephrase to reflect true extent of the study. Most of the assays were done on subsets of the samples, and many were done on very small subsets.

2. P12 line 241, change principle to principal.

3. The simple approach to subdivide the samples into NF2 and non-NF2 and carry out two separate comparisons between atypical and benign is sound (although a more rigorous ANOVA adjusting for NF1 status may have more power). It is assuring to see that there is a number of genes that are highly significant in both comparisons, suggesting that there are indeed expression differences between the atypical and benign groups. However, some of what follows is statistically misleading.

- Figure 2c makes it appear that after this subdivision analysis, atypical and benign samples can be clearly clustered into distinct classes. However, this result has little (if anything) to do with the subdivision for NF2 status. If one selects a set of genes differentially expressed between two cohorts, it is not surprising that the cohorts can be clustered using this set of genes. The same result can be obtained with an absolutely null dataset – and hence is not interesting. This in fact represents supervised clustering in disguise. This figure and associated figure legend is currently

misleading: "Heatmap of genes associated with atypical samples, which clearly separate them from benign meningiomas, is shown". The reasoning behind the clear separation is circular. This figure should be removed.

- Predictive model. "We next used 25 most differentially expressed genes to build a prediction algorithm... sufficient to classify 96% of samples into correct grade." I assume that a standard procedure for training and evaluating the random forest model has been used. Such approaches are now well established. However, by selecting 25 most differentially expressed genes – differentially expressed between the atypical and benign samples – the authors are biasing the procedure in their favor. It is not surprising that by selecting the most differentially expressed genes – based on the entire sample – the authors are able to classify bootstrap sub-samples. The statistical independence intrinsic to machine learning evaluation is removed. The resulting 96% prediction rate estimate is meaningless, and it is impossible to determine whether there is any prediction value at all. If the authors chose to pursue the model approach, it needs to be validated on an independent dataset, or some other, unbiased evaluation method must be applied. Otherwise the model must be removed from the manuscript.

- Figure 3b. I have the same objection as above. It is a nice visual separation, but is an expected effect of selecting the miRNAs that best separate the pre-selected classes.

- Page 13 line 270. "In order to correlate down-regulated miRNA clusters with gene expression patterns, we inferred miRNA:mRNA regulatory networks using samples that contained both datasets". This analysis makes sense, but it is difficult to assess the significance of the results, since correlations between expression levels naturally occur throughout the genome. As a control, the authors should compare the number of "negatively correlated targets" to positively correlated targets. If those numbers are similar, the analysis is likely spurious and should be removed. If those numbers are significantly different, this comparison should be quoted in the text as support for validity of the approach.

- Page 15. "Atypical tumors harbor hypermethylation of Polycomb Repressive Complex 2 (PRC2) binding sites". This is an interesting observation, but much of the discussion that follows is circular. The authors find that PRC2 and Homeobox sites are the most overrepresented group among differentially methylated regions (most of which are hypermethylated in atypical samples). This is interesting. But they follow this by stating "Interestingly we discovered a significant positive correlation between increased methylation of PRC2 and Homeobox with the likelihood of being an atypical meningioma". At this point, it is not interesting, it is just saying the same thing backwards. I was also curious as to what is meant by the "likelihood of being atypical". How is this likelihood determined?

- Line 357 "Based on the comparison of atypical versus benign meningiomas, we identified a significant increase in methylation at ESC EZH2 sites in atypical samples". This is not unexpected. Atypical are hypermethylated in general as compared to benign. Is the increase more significant at those sites than expected overall?? Even if it is, the authors have already shown that PRC2 targets (obtained from ESC data) are hypermethylated. Since EZH2 sites (in ESC data) are highly overlapping with PRC2 targets, it is not surprising that EZH2 binding sites will have similar profile. This whole section is made to appear very meaningful, but other than finding that PRC2/homeobox sites are the most highly overrepresented in hypermethylated sites, the rest of the section is simply a necessary outcome of the intercorrelations between the variables. None of the presented stats support the final conclusion that PRC2 methylation at those sites is mediated by increase in EZH2 – levels.

- P17, "Histone H3 lysine 27 tri-methylation (H3K27me3) chromatin immunoprecipitation sequencing (ChIP-seq) confirms silencing of PRC2 targets". It is an interesting finding, that K27me3 can be used to distinguish atypical vs. benign. However, it is not at all relevant that K27me3 changes are inversely correlated with gene expression changes. This will be true of any 2

sample types: fibroblasts vs lymphocytes, one cancer vs another, anything vs anything, as long as they are distinguishable at either expression or trimethylation levels. Trimethylation is associated with silencing, period. This is well known. This sounds meaningful but provides no insight into this comparison. Please remove.

- P18 line 380. "Furthermore, when we specifically focused on the differentially bound H3K27me3 regions and examined the PRC2-ESCs targets, we identified a significant increase in H3K27me3 binding of the PRC2-ESCs target regions in atypical samples as compared to benign ones". Again this is a direct consequence of the previous sentence (higher k27me3 binding in atypical). If it is higher genomewide, why would it not be higher at PRC2 targets? The interesting result would be to show that it is higher at those sites than expected from the overall increase – which is actually probably the case, since the K27me3 is generally concentrated around the set of developmentally relevant (PRC2 target genes), but the current statistic does not test for this. Also, this would be quite difficult to show convincingly, since at those sites PRC2 target sites have highest signal to noise ratio, and will always be the most prominent in a comparison.

- P18 "Given the role of epigenetic silencing in atypical samples, we wondered if activating epigenetic alterations may also be involved in formation of these tumors" – the analysis that follows in no ways tests/proves/supports the involvement of H3k27ac signal in the formation of atypical tumors. Any samples that display differential gene expression (perhaps with the exception of rapidly cycling or stimulated cells where chromatin does not have time to respond) will display differential promoter/enhancer usage, and hence differential K27ac marks. This analysis can be performed on any tissue type, any cancer, anything that is in some way different at gene expression level, and produce the same conclusion: "K27ac mark is associated with activation of gene expression." All results of this analysis are necessary outcomes of the above – well-established statement.

- Figure 4e. Are all of those comparisons necessary? The 4 bar plots simply reflect the fact that K27me3 targets, EED, Suz12, and PRC2 targets that have been deposited in the MSigDB are highly overlapping.

Reviewer #3 (Remarks to the Author): Expert in epigenomics

In this study Harnanci et al. performed a number of genomic, transcriptomic and epigenetic data to differentiated benign versus atypical meningiomas to understand whether the genomic landscapes of the two subtypes differ from each other. They report that atypical meningiomas show similar genomic features to benign tumors, including loss of NF2(although with different %), number of somatic mutation, same recurrent driver genes, same degree of chromosomal instability. The author claim that atypical tumors show distinct H3K27me3 and methylation profiles and upregulation of EZH2 as compared to benign meningiomas.

My major concerns about this work is that in spite of the effort the authors do not find substantial differences between benign and atypical meningiomas. The overall analysis is superficially done. While the study draws on a unique cohort to determine the architecture of atypical tumors, there are a number of conceptual and methodological issues that raise concerns about the interpretation of the data and the primary claims of the study. Furthermore, the authors stratify their cohort on the basis of alterations in NF2 and this makes more difficult to understand the overall differences (if any) between atypical and benign tumors. Clearly, tumors with loss of NF2 show different molecular features as compared to NF2 wild-type tumors regardless being atypical or benign. Moreover, epigenetic analyses are performed with a subgroup of tumors but the sample selection for these analyses are not specified.

Major comments

- The authors state that they performed targeted sequencing on a number of meningiomas. However, the experimental procedure that they used and the number of samples that underwent targeted sequencing are not present/specified. The authors must provide them. What is the average depth of coverage of the targeted sequenced samples?

- The authors identify somatic mutations from exome/target sequencing using the HaplotypeCaller software. This is somehow a not conventional choice of caller and its use is not recommended for somatic variant discovery as reported in the GATK software reference website and the use of MuTect is suggested. Thus, the results of variant calling should be compared with those generated by at least another caller such as MuTect. Likewise, the detection of small insertions/deletions should be compared to methods such as Strelka to avoid false positive calls. Hence, there are likely to be a number of artefacts.

- In the Methods section the authors do not specify what are the cutoffs used to identify somatic mutations rather they only list the filters. The authors must clearly state the cutoffs used.

-The authors look for somatic driver mutations but they do not report any details about these mutations. They only show information about significantly mutated "genes" (not mutations) across their samples as result of MutSig algorithm in Supp.Table 2. Information about somatic mutations present in each sample must be shown, especially if the authors want to stratify their cohort on mutations in genes of interest (as they claim in p. 7). Furthermore, the author must comment on the clonality of somatic mutations (i.e. the percentage of somatic cells that harbour the mutation). The author must distinguish between clonal and subclonal mutations. This distinction will clarify their stratification of samples. For instance, it is not clear whether a tumour with only subclonal (deleterious) mutations (e.g. mutations present in 5% of somatic cells) in NF2 is classified as an "NF2 loss" sample. Subclonal mutations (e.g. in 5% of somatic cells) acquired at very late stages of tumour evolution would not influence the feature of a tumour as clonal mutations (e.g. in 100% of somatic cells) that are acquired at the first stages of evolution. The author must clarify and comment on this point.

- To evaluate the accuracy (e.g. specificity) of variant calling procedure orthogonal validations (e.g. Sanger sequencing of a pool of randomly selected somatic mutations) must be performed.

- The authors compare the number of somatic coding mutations between atypical and benign samples and do not detect any differences (p.8 line 155). As a further validation of their variant calling, the mutation frequency of sequenced meningiomas should be compared to the one already reported in the literature (PMID: 23945592), or to other gliomas. It would be important to compare also the number of nonsilent somatic mutations (i.e. nonsynonymous, stopgain, stoploss, frameshift/nonframeshift insertion and deletion) per sequenced mega base pairs (mutation frequency) between atypical and benign melanomas.

- The authors find NF2 to be enriched for somatic mutations in atypical meningiomas as compared to benign tumours. Next they state that "we performed the same analysis in an extended cohort that included ...". It is not clear which is the additional cohort. The authors must clarify. In this second cohort the author find a higher number of somatic mutations in SMARCB1 (p.7 line 162). They conclude that on the basis of these observations "coding variation [...] does not significantly contribute to the risk of being atypical" (p.7 line 163). Did the author perform any statistical test/analysis to claim that somatic mutations do not "significantly" contribute to the risk of being atypical? Furthermore, this conclusion is opposite to the title of the paragraph where the authors state "coding variation in NF2 and SMARCB1 is associated with primary atypical samples". Which is the right conclusion? Is there any statistical evidence? It would also be interesting to comment whether mutations in NF2 co-occur with mutations in SMARCB1 (for instance, by performing comorbidity test).

- The authors stratify samples into two groups (i.e. high/low) on the basis of the mean percentage of genome alteration (PGA). No details about the distribution of the PGA across all samples are reported. As widely accepted, the mean of a distribution is affected by outliers (or extreme values) that can cause the mean to have a higher or lower value biased in favour of the direction of the outliers. This issue can influence the subsequent analyses and, most importantly, the conclusions. The author must report and comment on the distribution of the PGA. If the mean is biased by the outliers then a tumour with a PGA close to the mean PGA can erroneously be classified as CNV high/low. If the authors want to divide samples into two groups, the median is a better estimator. They must compare the mean with the median of the distribution. If the mean is significantly different from the median the author must use the median to stratify their cohort.

- The authors claim that atypical tumours have an mRNA expression signature that distinguishes them from benign meningiomas (p.11 line 218). This is a strong statement especially under the light of PCA analysis that clearly cannot separate atypical from benign tumours (Fig.2b and line 225). The authors present the results stratified by NF2-loss and this creates confusion in interpreting results. The authors must present the results of the PCA for atypical and benign tumours without any stratification, even as a supplementary figure. The results of the PCA should be shown as first (the axes should also report the percentage of variability explained by each component). As previously done, the authors did not clearly specify in the methods any cutoff that they used for the identification of differentially expressed genes. How do they define differentially expressed genes? How many differentially expressed genes do they find? Apparently, a table with gene expression values in each sample is not provided. Fig 2c is not clear and, presented as it is, is not informative. Fig 2d is shown as an example but it does not contain any relevant information (i.e. result of the analysis).

- The author uses a predictive algorithm (i.e. random forest) to classify meningiomas accordingly to the expression of 25 selected genes. Again, the authors do not provide any details about the implementation of this analysis in the Methods. The authors must provide these details. How do they create/train their model? Do they use the model on a test set? How do they evaluate the performance of their model? Reporting that 96% of samples were correctly identified is not sufficient. Moreover, the model is generated considering only the top 25 differentially expressed genes in atypical versus benign tumours. As a consequence, one can expect that this predictive algorithm will separate correctly the two groups. More importantly, to state that atypical tumours have a specific mRNA expression signature able to separate them from benign tumours, all differentially expressed genes should be included in the model. What are the results of this analysis if the author includes all differentially expressed genes?

- The authors analyze DNA methylation data (p.14) but they do not comment on whether atypical tumours differ from benign ones (which should be the topics of their paper as stated in p.3 line 38). It should be worth commenting on this point.

Minor comments

- The proportion of atypical tumours in p.5 line 88 (i.e. 10%) is different from the estimates reported in p. 4 line 66 (i.e. 5-20%). Which is the real fraction of atypical meningiomas?
- Supplementary Table 1a and 1b must be merged since they are redundant.
- Colour keys reported in Figures are sometimes redundant. Please reduce this redundancy. Sometimes the same colour is used in different figures to indicate distinct features. This creates confusion. Colours must be consistent across figures. A colour must be used to indicate univocally only one feature among all figures.
- At line 184 (p.9) the author rounded the p-value present in figure 1c (0.27) to 0.2. The correct rounding is 0.3.
- The authors show that losses of chr14q, 10q, 10p, 1p and 6q are significantly different between atypical and benign meningiomas (p.10 line 195). The authors report the FDR values but they must also provide the p values of the Fisher tests.

We thank the reviewers for their insightful comments and criticisms. Based on their input, we have performed several additional experiments and analyses. These are detailed below along with our point-by-point response to their valuable comments.

Reviewer #1 (Remarks to the Author):

The aim of the current study was to dissect molecular features defining primary de novo atypical (grade II) meningiomas. They performed next generation or targeted sequencing of 775 meningiomas, including 468 primary grade I and 88 primary atypical meningiomas. 75% of the grade II tumors contained NF2 mutations and 9% TRAF7/PI3K mutations. Targeted sequencing revealed a fraction of NF -mutated atypical meningiomas with simultaneous SMARCB1 mutations. They conclude that NF2/SMARCB1 mutation defines atypical phenotype. Moreover, primary NF2-mutated grade II meningiomas were characterized by high copy number variation (CNV). NF2 mutant tumors have greater chromosomal instability. Another feature of atypical tumors was chromosome 14q loss. Specific genes associated with chromosomal losses were PTEN, NOTCH2, MAX, and few others. Analyzing the mRNA profile the authors found that atypical tumors have upregulation of E2F and FOXM1 transcription factor networks. NF2-mutated and non-mutated meningiomas had a different miRNA profile, with let 7 family members being prominent. Analysis of the DNA methylation status from 60 samples revealed atypical meningioma subgroups. Another interesting finding was PRC2 hypermethylation in atypical and benign high-CNV meningiomas.

They also studied a subgroup of progressive meningiomas. Paired samples from 4 patients (all NF2 mutant).

In primary, non-recurrent meningiomas, no TERT promoter mutation was detected. The authors propose two different pathways for the development of atypical meningiomas - either de novo or due to transformation of primary benign meningiomas through TERT promoter mutations.

Comments:

The authors performed a comprehensive study using an impressive number of samples. However, a fraction of their observations has been already described before, including the genetic instability of NF2-mutated tumors, the NF2-SMARCB1 association, the chromosomal losses, and some expression data. Thus, some results have more confirmatory character than being new.

We thank the reviewer for their insightful comments about the manuscript. In our previous study we performed whole genome genotyping and whole-exome sequencing on 8 atypical and 42 benign tumors¹, however H3K27ac ChIP-Seq and microarray gene expression analysis did not include any primary atypical samples. Similarly, the study that we recently published on somatic *POLR2A* mutations

focuses only on benign meningiomas². On the contrary, in this study, we have analyzed substantially more atypical meningiomas, including whole exome sequencing (n=10), whole-genome genotyping (n=55), gene expression (n=26) and H3K27ac ChIP-Seq (n=3). Additionally, for the first time we have performed DNA methylation analysis of 57 tumors (n=11 atypical, n=46 benign), miRNA expression profiling on 32 tumors (n=16 atypical, n=16 benign), and H3K27me3 ChIP-Seq on 6 tumors (n=3 benign, n=3 atypical). Moreover, we performed whole-exome sequencing on 4 paired meningioma samples that included the original benign sample as well as the progressed atypical counterpart from the same patient. While we agree with the reviewer that the genetic instability of *NF2* mutated tumors and *NF2-SMARCB1* co-mutation has been described in our previous studies, the genomic and epigenomic landscape of atypical samples has not yet been systematically studied. We hope that the reviewer would agree this is the first study in the literature that comprehensively and specifically investigates the molecular mechanisms of atypical meningiomas.

Some questions remain:

1) We know from clinical experience that atypical meningiomas can be very different in clinical course. So it would be highly interesting (and would add clinical relevance to the manuscript) to relate some features (CNV, others) to the time to tumor recurrence or the presence/absence of atypical meningioma recurrence. We see a fraction of atypical meningiomas which have no aggressive biology at all.

We thank the reviewer for this insightful comment. Unfortunately, extensive clinical annotation is not available for our entire dataset, however, we have analyzed the available data and made some interesting preliminary findings that will need to be studied in larger clinical sample sets. For example, considering the time to recurrence and CNV information in three samples, we observed that the time to recurrence and percentage of genome alteration (PGA) was inversely correlated. The sample with the shortest time to recurrence interval had the highest PGA value (recurrence: 45 months, PGA: 12%) whereas the sample with the longest time revealed the reverse (recurrence: 144 months, PGA: 0%) (Supplementary Table 1a).

To further respond to the reviewer's excellent suggestion, we also studied the paired meningioma samples that included the original primary benign sample as well as the progressed atypical counterpart from the same patient. When we compared the CNV profile of progressed atypical tumor with their corresponding benign counterpart, we identified large-scale deletions in 9p, 14q and LOH in chr3 (Supplementary Table 11c). We also compared recurrent atypical meningiomas to primary atypical ones and analyzed the whole genome genotyping data (n=5 recurrent atypical samples). All atypical recurrent samples were CNV-high. The most observed deletion events in recurrent atypical samples were similar to those in primary atypical samples (1p (n=5), 22q (n=4), 14q (n=2), 6q (n=2), 2p (n=2), 7p(n=2), 9p(n=2) and 18q (n=2)). We next examined DNA methylation data of

recurrent atypical samples (n=3). Recurrent atypical meningiomas clustered with primary atypical meningiomas based on DNA methylation profile. Similar to primary atypical samples, atypical recurrent samples also revealed the hypermethylated phenotype, especially hypermethylation of the ESC-PRC2 target sites. When we considered differentially methylated sites between atypical recurrent and primary samples, we identified only one probe that was significant, and it fell in the promoter region of *ETV5* gene. Lastly, we studied microarray gene expression data of 3 recurrent atypical samples. Similar to primary atypical samples, recurrent atypical samples also showed up-regulation of the *EZH2* and *E2F2/FOXM1* transcriptional networks. When we compared atypical recurrent (n=3) to atypical primary samples, we identified only 5 genes (*FAM83D*, *LGR4*, *ARSI*, *POSTN*, *TRIM54*) that were differentially expressed. We have now included this data as the new Supplementary Table 11d and cited it in the main text of our manuscript.

2) Are there any NF2 patients in the group?

We thank the reviewer for this comment. In our study, we excluded patients with germline *NF2* mutations. We confirmed this by re-analyzing the whole exome sequencing data of the blood samples and indeed did not identify any patients with inherited or de novo germline *NF2* mutations.

3) At least a few cases with multiple (atypical) meningiomas should be studied.

Our meningioma cohort included 5 cases with multiple meningiomas, including one patient with an atypical tumor. We performed targeted sequencing of a single tumor from each patient, identifying somatic *NF2* mutations in 3 cases, including the atypical meningioma. The remaining 2 tumors did not harbor mutations in known meningioma drivers. We have added a column to Supplementary Table 1a detailing which samples were obtained from patients with multiple meningiomas.

Additionally, we performed whole-exome sequencing of 4 different samples from one large atypical meningioma tumor. We detected somatic *NF2* mutations in all of the samples and a *SUFU* mutation in 3 out of 4 samples. We calculated the clonality rate of these mutations based on the variant allele frequency and ploidy at that site, taking into account the admixture rate of each tumor as described previously³. The clonality rates of *NF2* and *SUFU* mutations were nearly 100% in the samples that harbored these variants, suggesting that both *SUFU* and *NF2* mutations were highly penetrant in the regions they were found and are likely biologically meaningful. Further studies will be needed to study additional patients with multiple meningiomas.

4) Atypical meningiomas are usually treated by irradiation after surgery. It would be interesting to compare non-irradiated vs. irradiated atypical tumor to see if a certain molecular feature predicts the need for irradiation.

We agree with the reviewer that identifying the molecular features that can predict the need for irradiation would be very beneficial. Unfortunately, we do not have before and after radiation samples available from the same patient. However, we agree with the reviewer that this will be an interesting clinical question to pursue.

5) The histology of the “benign” tumors should be taken into account – how close are the fibrous grade I with NF2 mutation to atypical tumors (with/without NF2 mutation)?

We thank the reviewer for this comment. In our 556 meningioma cohort, we have 36 benign fibrous samples. Of these, 31 are *NF2* mutant. When we compared benign fibrous *NF2* samples to atypical tumors, we found that primary fibrous atypical tumors were significantly more likely to be CNV-high, as compared to fibrous benign ones ($P= 0.004$). Considering the gene expression data, we again observed overexpression of *EZH2* in fibrous atypical tumors as compared to fibrous benign ones ($FDR=0.03$). Gene expression PCA plot using the mRNA signature genes successfully distinguished fibrous benign *NF2* meningiomas from atypical ones (Supplementary Figure S9). Similarly, in DNA methylation, miRNA-Seq and H3K27ac Chip-Seq datasets, fibrous benign *NF2* samples clustered together with other benign tumors, and not with the atypical samples.

Updated Supplementary Figure S9: Gene expression PCA plot using the mRNA signature genes distinguishes fibrous benign *NF2* mutant meningiomas from atypical ones. Different histology types are color coded, which is shown at the bottom.

Minor comments

1) Ref.2: The 2007 WHO classification is no longer valid because a new classification has been published recently

Thank you for bringing this to our attention. We have now corrected the reference to include the most recent WHO classification (new manuscript lines 65 and 69).

2) what is known regarding the let-7 miRNA family in meningiomas?

A previous study suggested that let-7 miRNAs might play a role in the benign to malignant transition stages of schwannomas and extended this finding to

meningiomas⁴, without detailing any specific results. To our knowledge, ours is the largest study analyzing the differential miRNA expression in atypical versus benign meningiomas, documenting a clear difference.

3) supplementary figure 2 has some strange characters above the figure

We apologize that we did not make this clear. We tried to summarize the EZH2 staining quantifications using symbols above the figures. We have now corrected this in our updated Supplementary Figure S13.

Reviewer #2 (Remarks to the Author): Expert in brain cancer genomics

The authors carry out genomic/epigenomic analyses on subsets of a large set of meningiomas. They find that atypical (malignant) tumors are characterized by NF2 mutations, genomic instability, CpG hypermethylation, and increased K27me3 levels.

The above results are interesting and appear solid. However, many of the analyses presented in the manuscript do not provide additional support to the role of epigenetic reprogramming and many of the presented results are simple necessary outcomes of well-known relationships between chromatin states and transcription. The manuscript needs to clearly distinguish between primary findings (as enumerated in the paragraph above) and descriptive analyses, such as e.g. finding downregulated genes to be associated with increased K27me3, which is well known, and will be true in any comparison.

Most of the abstract and discussion are OK, but a substantial part of the result section needs to be toned down/removed to clearly indicate which analyses are as consistent with general known relationships between chromatin and expression, as they are with involvement of chromatin modifications in meningioma.

We thank the reviewer for constructive assessment of our manuscript. We have adjusted the wording in the result section to emphasize above points in the revised manuscript.

1) Abstract “Based on comprehensive genomics, transcriptomics and epigenomic analyses of 775 meningiomas” – This is misleading, please rephrase to reflect true extent of the study. Most of the assays were done on subsets of the samples, and many were done on very small subsets.

We appreciate the reviewer’s input and have now rephrased this sentence to read “Based on comprehensive genomics, transcriptomics and epigenomic analyses of meningiomas” in new text line 38 of the manuscript.

2) P12 line 241, change principle to principal.

Thank you for bringing this to our attention. We have now corrected *principle* to *principal* in new text line 322 of the manuscript.

3) The simple approach to subdivide the samples into NF2 and non-NF2 and carry out two separate comparisons between atypical and benign is sound (although a more rigorous ANOVA adjusting for NF1 status may have more power). It is assuring to see that there is a number of genes that are highly significant in both comparisons, suggesting that there are indeed expression differences between the atypical and benign groups. However, some of what follows is statistically misleading. Figure 2c makes it appear that after this subdivision analysis, atypical and benign samples can be clearly clustered into distinct classes. However, this result has little (if anything) to do with the subdivision for Nf2 status. If one selects a set of genes differentially expressed between two cohorts, it is not surprising that the cohorts can be clustered using this set of genes. The same result can be obtained with an absolutely null dataset – and hence is not interesting. This in fact represents supervised clustering in disguise. This figure and associated figure legend is currently misleading: “Heatmap of genes associated with atypical samples, which clearly separate them from benign meningiomas, is shown”. The reasoning behind the clear separation is circular. This figure should be removed.

We appreciate the reviewer’s input. We initially included this figure as it visually confirms the separation strength of the signature genes to be robust. However, we agree that this presentation of the clustering of samples based on signature genes is biased and based on the reviewer’s input, we have now removed the figure and the associated figure legend in the revised manuscript.

In response to reviewer’s comment about stratifying the cohort on the basis of *NF2* status, since we observed a significant association of *NF2* mutations with atypical tumors, we thought that stratifying the cohort on the basis of *NF2* status was needed. We considered that a simple comparison between benign and atypical samples would largely be driven in large part by underlying driver mutation and aimed at accounting for this observation.

Predictive model. “We next used 25 most differentially expressed genes to build a prediction algorithm... sufficient to classify 96% of samples into correct grade.” I assume that a standard procedure for training and evaluating the random forest model has been used. Such approaches are now well established. However, by selecting 25 most differentially expressed genes – differentially expressed between the atypical and benign samples – the authors are biasing the procedure in their favor. It is not surprising that by selecting the most differentially expressed genes – based on the entire sample – the authors are able to classify bootstrap sub-samples. The statistical independence intrinsic to machine learning evaluation is removed. The resulting 96% prediction rate estimate is meaningless, and it is impossible to

determine whether there is any prediction value at all. If the authors chose to pursue the model approach, it needs to be validated on an independent dataset, or some other, unbiased evaluation method must be applied. Otherwise the model must be removed from the manuscript.

The reviewer raises an important point. We agree with the reviewer's concern that validating our prediction model on the training set creates a bias. To address this issue, we have now validated our prediction model on an independent meningioma gene expression dataset⁵. Even though our validation cohort was generated using Affymetrix platform, which is different than the platform (Illumina) of our training cohort, we correctly predicted 42/43 benign samples. When we considered histologically atypical meningiomas with high and medium Ki-67 index, our algorithm correctly predicted 6/9 samples as atypical. Taken together with 42/43 benign samples which were correctly identified, our prediction accuracy for this replication cohort was 91%.

Based on the reviewer's input, we also changed our wording in the text to emphasize that 96% prediction accuracy (4% error rate) corresponds to an 'out of bag' (OOB) error rate. We believe that reporting OOB is still justified because OOB error is computed using cross-validation in the training step of random forest. By definition, OOB in the random forest model represents the error rate of the trained algorithm on a leftout dataset that is not used in the training algorithm. Therefore it reflects the generalized testing error of the trained model.

We have adjusted the wording to present above findings in the revised manuscript (in new text line 255 and 668).

4) Figure 3b. I have the same objection as above. It is a nice visual separation, but is an expected effect of selecting the miRNAs that best separate the pre-selected classes.

We appreciate the reviewer's comment. We have now included the names of differentially expressed miRNAs in Figure 3b, which can be used to visually inspect the expression profile of each miRNA across all samples. We hope that the reviewer agrees that the updated version of the Figure 3b is informative.

5) Page 13 line 270. "In order to correlate down-regulated miRNA clusters with gene expression patterns, we inferred miRNA:mRNA regulatory networks using samples that contained both datasets". This analysis makes sense, but it is difficult to assess the significance of the results, since correlations between expression levels naturally occur throughout the genome. As a control, the authors should compare the number of "negatively correlated targets" to positively correlated targets. If those numbers are similar, the analysis is likely spurious and should be removed. If those numbers are significantly different, this comparison should be quoted in the text as support for validity of the approach.

We thank the reviewer for this very insightful comment. We agree that comparing the number of negatively correlated targets to positively correlated targets would be useful in showing the validity of our approach. In response to this suggestion, we calculated the number of negatively and positively correlated targets of each miRNA. We identified that the number of negatively correlated targets to be significantly higher than positively correlated targets ($P < 2.2e-16$, paired Wilcoxon test), which we believe confirms the validity of our results.

This result is shown below and included as the new Supplementary Figure S11. We also adjusted the wording to present above findings in our revised manuscript (in new text line 295).

Updated Supplementary Figure S11. Barplots of the number of negatively and positively correlated targets of miRNAs (negative:blue, positive:red). The number of negatively and positively correlated targets of miRNAs is statistically different ($P < 2.2e-16$). Wilcoxon paired test was used to calculate significance.

6) Page 15. “Atypical tumors harbor hypermethylation of Polycomb Repressive Complex 2 (PRC2) binding sites”. This is an interesting observation, but much of the discussion that follows is circular. The authors find that PRC2 and Homeobox sites are the most overrepresented group among differentially methylated regions (most of which are hypermethylated in atypical samples). This is interesting. But they follow this by stating “Interestingly we discovered a significant positive correlation between increased methylation of PRC2 and Homeobox with the likelihood of being an atypical meningioma”. At this point, it is not interesting, it is just saying the same thing backwards. I was also curious as to what is meant by the “likelihood of being atypical”. How is this likelihood determined?

We agree with the reviewer’s comment that the above sentence is not necessary. We have now removed the following sentence in the revised manuscript: “Importantly we discovered a significant positive correlation between increased methylation of PRC2 and Homeobox with the likelihood of being an atypical meningioma”.

The reviewer also asked how the likelihood of being atypical is calculated. We subtracted the averaged methylation values of control samples from each tumor sample. We next calculated the number of hypermethylated Homeobox and ESC-PRC2 binding target sites. We then built two different logistic regression models to predict the atypical/benign variable either from the number of hypermethylated PRC2 target sites or from the number of hypermethylated Homeobox target size. We considered the odds ratio calculated from the logistic regression to represent the constant effect of the predictor (i.e. number of hypermethylated PRC2 target sites/number of hypermethylated Homeobox targets) on the likelihood of being atypical. In response to the reviewer’s suggestion we removed this analysis from the revised manuscript.

7) Line 357 “Based on the comparison of atypical versus benign meningiomas, we identified a significant increase in methylation at ESC EZH2 sites in atypical samples”. This is not unexpected. Atypical are hypermethylated in general as compared to benign. Is the increase more significant at those sites than expected overall?? Even if it is, the authors have already shown that PRC2 targets (obtained from ESC data) are hypermethylated. Since EZH2 sites (in ESC data) are highly overlapping with PRC2 targets, it is not surprising that EZH2 binding sites will have similar profile. This whole section is made to appear very meaningful, but other than finding that PRC2/homeobox sites are the most highly overrepresented in hypermethylated sites, the rest of the section is simply a necessary outcome of the intercorrelations between the variables. None of the presented stats support the final conclusion that PRC2 methylation at those sites is mediated by increase in EZH2 – levels.

We thank the reviewer for this comment. We found that among the hypermethylated sites in atypical samples (which indeed were broader and deeper than those in benign samples), there was particular enrichment for the ESC-EZH2

target binding sites ($P < 2.2e-16$, Fisher's Exact test). The increase at the ESC-EZH2 binding sites was significantly more than expected by chance.

We agree with the reviewer that the ESC-EZH2 sites highly overlap with the ESC-PRC2 targets. ESC-PRC2 binding sites were generated using ChIP-on-chip whereas the ESC-EZH2 binding sites were generated by the ENCODE consortium using ChIP-Seq data. We included these two different datasets generated by two different approaches in order to validate and reproduce our observations. However, we appreciate the reviewer's comment and based on this suggestion, we moved the ESC-EZH2 Figure 4f to supplementary Figure S12.

We also agree with the reviewer that we did not conclusively prove EZH2 overexpression to be the mediator of ESC-PRC2 binding site hypermethylation. We presented independent data that of EZH2 overexpression as well as increased methylation of ESC-PRC2 target sites in the atypical tumors. We followed the previous literature which has shown EZH2, the catalytic subunit of PRC2, to be a recruitment platform for DNA methyltransferases (DNMTs), acting as a direct controller of DNA methylation at PRC2 binding sites⁶. Based on this knowledge and our observations, we suggested the association between increased methylation in atypical tumors may be related to deregulated PRC2 activity, as observed in various cancer types^{7,8}. We now changed the wording in the revised manuscript, emphasizing that this association was not experimentally tested in our study (new text line 372).

8) P17, "Histone H3 lysine 27 tri-methylation (H3K27me3) chromatin immunoprecipitation sequencing (ChIP-seq) confirms silencing of PRC2 targets". It is an interesting finding, that K27me3 can be used to distinguish atypical vs. benign. However, it is not at all relevant that k27me3 changes are inversely correlated with gene expression changes. This will be true of any 2 sample types: fibroblasts vs lymphocytes, one cancer vs another, anything vs anything, as long as they are distinguishable at either expression or trimethylation levels. Trimethylation is associated with silencing, period. This is well known. This sounds meaningful but provides no insight into this comparison. Please remove.

We appreciate the reviewer's point. Based on this suggestion, we removed the following sentence in the revised manuscript: "Indeed, we were able to identify a significant negative correlation between the H3K27me3 signal and changes in gene expression levels in atypical versus benign samples (~ -0.12 , $< 2.2 \cdot 10^{-16}$ correlation test)".

9) P18 line 380. "Furthermore, when we specifically focused on the differentially bound H3K27me3 regions and examined the PRC2-ESCs targets, we identified a significant increase in H3K27me3 binding of the PRC2-ESCs target regions in atypical samples as compared to benign ones". Again this is a direct consequence of the previous sentence (higher k27me3 binding in atypical). If it is higher genomewide, why

would it not be higher at PRC2 targets? The interesting result would be to show that it is higher at those sites than expected from the overall increase – which is actually probably the case, since the K27me3 is generally concentrated around the set of developmentally relevant (PRC2 target genes), but the current statistic does not test for this. Also, this would be quite difficult to show convincingly, since at those sites PRC2 target sites have highest signal to noise ratio, and will always be the most prominent in a comparison.

We thank the reviewer for this comment. We performed gene set enrichment analysis on regions with increased H3K27me3 binding and decreased expression. The GREAT algorithm computes ontology term enrichments using a hypergeometric test that assesses if the enrichment for any given ontology term is more than expected by chance. Based on the GREAT enrichment analysis, we identified the ESC-PRC2 target binding sites to have significantly more H3K27me3 signal and decreased expression in atypical meningiomas as compared to benign ones (FDR= 0.03), showing that the increase at ESC-PRC2 binding sites was significantly more than expected by chance. Put another way, in the context of increased H3K27me3 binding in atypical samples, the regions that were differentially bound were particularly enriched for ESC-PRC2 binding sites. Since GREAT analysis assesses the enrichment using hypergeometric test, the enrichment was significant even though there was an overall increase in H3K27me3 signal in atypicals. We believe this finding has implications in the pathology of atypical meningiomas. We presented the above findings in our revised manuscript (in new text line 399).

10) P18 “Given the role of epigenetic silencing in atypical samples, we wondered if activating epigenetic alterations may also be involved in formation of these tumors” – the analysis that follows in no ways tests/proves/supports the involvement of H3k27ac signal in the formation of atypical tumors. Any samples that display differential gene expression (perhaps with the exception of rapidly cycling or stimulated cells where chromatin does not have time to respond) will display differential promoter/enhancer usage, and hence differential K27ac marks. This analysis can be performed on any tissue type, any cancer, anything that is in some way different at gene expression level, and produce the same conclusion: “K27ac mark is associated with activation of gene expression.” All results of this analysis are necessary outcomes of the above – well-established statement.

We appreciate the reviewer’s comment. We agree with the reviewer that we did not conclusively establish the involvement of activating epigenetic alterations in the formation of atypical tumors. We have now revised the text to read “Given the role of epigenetic silencing in atypical samples, we investigated if activating epigenetic alterations may also be **associated** with these tumors” (new text line 407).

While we acknowledge that any two sample populations with differential gene expression will consequently have differential promoter/enhancer usage, we believe our analysis of H3K27ac signal to identify differential super enhancers in

atypical tumors offers novel insights. Identification of these super-enhancer elements has been shown to be informative in elucidating molecular changes associated with various cancer types⁹. We hope that the reviewer would agree with our belief that this new insight, which we obtained through super-enhancer analysis, provides novel understanding into epigenetic changes that are associated with these atypical tumors.

11) Figure 4e. Are all of those comparisons necessary? The 4 bar plots simply reflect the fact that K27me3 targets, EED, Suz12, and PRC2 targets that have been deposited in the MSigDB are highly overlapping.

We appreciate the reviewer's input. We agree with the reviewer that ESC-H3K27me3, ESC-EED, ESC-SUZ12 and ESC-PRC2 targets are highly overlapping. We have now updated Figure 4e to include only the ESC-H3K27me3 and ESC-PRC2 targets in accordance to the reviewer's comments.

Reviewer #3 (Remarks to the Author): Expert in epigenomics

In this study Harnanci et al. performed a number of genomic, transcriptomic and epigenetic data to differentiated benign versus atypical meningiomas to understand whether the genomic landscapes of the two subtypes differ from each other. They report that atypical meningiomas show similar genomic features to benign tumors, including loss of NF2(although with different %), number of somatic mutation, same recurrent driver genes, same degree of chromosomal instability. The author claim that atypical tumors show distinct H3K27me3 and methylation profiles and upregulation of EZH2 as compared to benign meningiomas.

My major concerns about this work is that in spite of the effort the authors do not find substantial differences between benign and atypical meningiomas. The overall analysis is superficially done. While the study draws on a unique cohort to determine the architecture of atypical tumors, there are a number of conceptual and methodological issues that raise concerns about the interpretation of the data and the primary claims of the study. Furthermore, the authors stratify their cohort on the basis of alterations in NF2 and this makes more difficult to understand the overall differences (if any) between atypical and benign tumors. Clearly, tumors with loss of NF2 show different molecular features as compared to NF2 wild-type tumors regardless being atypical or benign. Moreover, epigenetic analyses are performed with a subgroup of tumors but the sample selection for these analyses are not specified.

We thank the reviewer for constructive assessment of our manuscript.

We attempted to carefully design all our experiments to disentangle the confounding effect of CNVs and underlying mutational profiles (including loss of

NF2). This decision was made due to the significant association of *NF2* loss with atypical tumors and our concern that a simple comparison between benign and atypical samples could be largely driven by this underlying driver mutation. Regarding our epigenetic analyses, we selected DNA methylation and miRNA expression samples with overlapping microarray gene expression data. In contrast to studies attempting to derive epigenetic information from artificially cultured tumor lines (~100 million cells), H3K27ac/H3K27me3 ChIP-Seq was performed on fresh-frozen primary tumor or dura specimens (~100,000 cells). The stringent quality requirements for performing ChIP-Seq on fresh-frozen tumor sections derived from patients in the operating room fundamentally limited the number of tumor samples eligible for ChIP-seq. Among the eligible ones, we selected representative, well-controlled samples from each subgroup for ChIP-Seq. In addition, we considered the ChIP-Seq samples in junction with overlapping gene expression and DNA methylation data in order to correlate these orthogonal datasets to gain unique and valid insight into pathogenesis of these tumors.

Major comments

1) The authors state that they performed targeted sequencing on a number of meningiomas. However, the experimental procedure that they used and the number of samples that underwent targeted sequencing are not present/specified. The authors must provide them. What is the average depth of coverage of the targeted sequenced samples?

We thank the reviewer for this important point. We detailed this information in Supplementary Table 1b, which documents the type of sequencing used (whole-exome, targeted next-generation sequencing, and/or Sanger sequencing). Our average depth of coverage of the targeted sequenced samples was around 2000x. As suggested by the reviewer, in our revision, we now present the depth of coverage of each targeted sequenced samples in Supplementary Table 1b.

2) The authors identify somatic mutations from exome/target sequencing using the HaplotypeCaller software. This is somehow a not conventional choice of caller and its use is not recommended for somatic variant discovery as reported in the GATK software reference website and the use of MuTect is suggested. Thus, the results of variant calling should be compared with those generated by at least another caller such as MuTect. Likewise, the detection of small insertions/deletions should be compared to methods such as Strelka to avoid false positive calls. Hence, there are likely to be a number of artefacts.

We appreciate the reviewer's input. Based on this suggestion, we have now compared the list of our somatic point mutation and indel calls with that created by using the MuTect¹⁰/ Indelocator (<http://archive.broadinstitute.org/cancer/cga/indelocator>) as well as Strelka¹¹ variant callers. Since MuTect is designed to detect somatic point mutations only, we

used it in conjunction with the short indel variant calling method, Indelocator. The median number of coding non-synonymous mutations identified using HaplotypeCaller, MuTect/Indelocator and Strelka is 15, 14 and 15, respectively. These mutation rates are very similar to previous meningioma study, which revealed an average rate of ~ 10 non-synonymous mutations per tumor¹².

In order to perform a systematic comparison of variant calling methods, we used the Sanger-confirmed driver mutations in our meningioma cohort as a 'gold standard' set. Interestingly, using stringent thresholds GATK HaplotypeCaller and Strelka did not identify the known driver mutations in 4 samples whereas the MuTect+Indelocator missed the previously identified driver mutations in 3 samples.

Importantly, as mentioned above, we Sanger confirmed all driver mutations presented in the manuscript, which ensures the specificity of our mutation calls. To ensure the sensitivity and as suggested by the reviewer, we now analyzed our data with three different calling methods and included these results in the revised manuscript (Supplementary Table 2c-e).

3) In the Methods section the authors do not specify what are the cutoffs used to identity somatic mutations rather they only list the filters. The authors must clearly state the cutoffs used.

We apologize for this oversight and agree with the reviewer that the cutoffs need to be clarified. We used the following quality metrics to filter out variants: (1) somatic score less than 20; (2) overlapping a RepeatMasker or segmental duplication annotated region; (3) low quality (< 30) and low quality-by-depth values (<1); (4) with many mapping quality zero reads; (5) with strand bias; (6) with mutation cluster of size more than 2; (7) homopolymer runs of length more than 10 base pairs within +/- 5 base pairs around the mutation or from the right of the mutation, and; (8) ClippingRankSum (calculated by GATK) less than -3.0 or more than 3.0 as described previously^{2,13}. To clearly state these criteria, we revised Methods section in new text line 588.

4) The authors look for somatic driver mutations but they do not report any details about these mutations. They only show information about significantly mutated "genes" (not mutations) across their samples as result of MutSig algorithm in Supp.Table 2. Information about somatic mutations present in each sample must be shown, especially if the authors want to stratify their cohort on mutations in genes of interest (as they claim in p. 7). Furthermore, the author must comment on the clonality of somatic mutations (i.e. the percentage of somatic cells that harbour the mutation). The author must distinguish between clonal and subclonal mutations. This distinction will clarify their stratification of samples. For instance, it is not clear whether a tumour with only subclonal (deleterious) mutations (e.g. mutations present in 5% of somatic cells) in NF2 is classified as an "NF2 loss" sample. Subclonal mutations (e.g. in 5% of somatic cells) acquired at very late stages of tumour evolution would not

influence the feature of a tumour as clonal mutations (e.g. in 100% of somatic cells) that are acquired at the first stages of evolution. The author must clarify and comment on this point.

We thank the reviewer for these suggestions. We have now included an additional Supplementary Table 2c-e to list all the requested details of our somatic variant calls.

The reviewer raises an important issue about the clonality of the somatic mutations. Based on the reviewer's suggestion, we now report the calculated clonality rate of each somatic mutation based on the variant allele frequency and ploidy at that site, taking into account the admixture rate of each tumor as described previously³. For example, in the majority of our samples, we calculated *NF2* mutations to have a clonality rate of 100%, suggesting that *NF2* mutations indeed occurred early during tumor formation (Supplementary Table 2f, Supplementary Figure S4).

In the revised manuscript, we included a new Supplementary Figure S4 to report these findings. We also included new wording to present above findings in the revised manuscript (in new text line 170).

Updated Supplementary Figure S4: Clonality rates of *NF2* mutant atypical and benign tumors.

5) To evaluate the accuracy (e.g. specificity) of variant calling procedure orthogonal validations (e.g. Sanger sequencing of a pool of randomly selected somatic mutations) must be performed.

We thank the reviewer for this comment. We used Sanger sequencing to validate all of the driver mutations identified from whole-exome sequencing data.

6) The authors compare the number of somatic coding mutations between atypical and benign samples and do not detect any differences (p.8 line 155). As a further validation of their variant calling, the mutation frequency of sequenced meningiomas should be compared to the one already reported in the literature (PMID: 23945592), or to other gliomas. It would be important to compare also the number of nonsilent somatic mutations (i.e. nonsynonymous, stopgain, stoploss, frameshift/nonframeshift insertion and deletion) per sequenced mega base pairs (mutation frequency) between atypical and benign melanomas.

We thank the reviewer for this remark. Based on this suggestion, we formally compared the mutation frequency of meningioma to other cancer types¹⁴. As expected, the mutation frequency (median: 0.27) of meningioma is lower than malignant brain cancers, like glioblastoma. Finally, when we compared the mutation frequency of non-silent somatic mutations between atypical and benign meningiomas, we could not identify any statistically significant difference. We added Supplementary Figure S2 and S3 to show these findings.

Updated Supplementary Figure S2: Mutation frequency of meningiomas. Every dot represents a sample while the dashed horizontal lines are the median numbers of mutations (salmon:benign, blue:atypical). The median mutation frequency (median: 0.27) of meningioma is lower than glioblastoma¹⁴.

Figure From Alexandrov L.B. et al., Nature (2013)¹⁴.

Updated Supplementary Figure S3: Boxplots of the number of somatic indel, missense, stop-gain, top-loss mutation counts from 75 meningioma exomes normalized per millibase of sequencing (n=10 atypical, salmon; n= 65 benign, blue). The number of indel, missense, stop-gain and stop-loss somatic mutations from whole-exome sequencing data is not statistically different between atypical and other meningioma samples (P =0.71, indel; P = 0.37, missense, P =0.18, stop-gain, P = 0.34, stop-loss; Student’s t-test).

7) The authors find NF2 to be enriched for somatic mutations in atypical meningiomas as compared to benign tumours. Next they state that “we performed the same analysis in an extended cohort that included ...”. It is not clear which is the additional cohort. The authors must clarify. In this second cohort the author find a higher number of somatic mutations in SMARCB1 (p.7 line 162). They conclude that on the basis of these observations “coding variation [...] does not significantly contribute to the risk of being atypical” (p.7 line 163). Did the author perform any statistical test/analysis to claim that somatic mutations do not “significantly” contribute to the risk of being atypical? Furthermore, this conclusion is opposite to the title of the

paragraph where the authors state “coding variation in *NF2* and *SMARCB1* is associated with primary atypical samples”. Which is the right conclusion? Is there any statistical evidence? It would also be interesting to comment whether mutations in *NF2* co-occur with mutations in *SMARCB1* (for instance, by performing comorbidity test).

We apologize for not being clear and agree with the reviewer that above sentences need to be clarified in the main text. We first identified significantly mutated mutations among all atypical samples that underwent whole-exome sequencing using the MutSig algorithm. *NF2* was the only somatic mutation that we found to be enriched in atypical samples. We next studied a combined cohort that included not only the same whole-exome dataset, but also a new dataset of primary meningioma samples that were subjected to targeted sequencing (n=556). We calculated the atypical association significance of our driver mutations (*NF2*, *SMARCB1*, *TRAF7/PI3K*, *TRAF7/KLF4*, *POLR2A*, Hedgehog) using two-sided Fisher’s exact test. We observed significant atypical association only for *NF2* and *SMARCB1* mutations (*NF2*: $P= 2.2 \times 10^{-10}$, *SMARCB1*: $P=0.05$, Fisher’s exact test). A tumor harboring *NF2* loss was 3.78 times more likely to be atypical compared to a non-*NF2* meningioma. Similarly, a tumor harboring *NF2/SMARCB1* mutation had a 2.96 times greater risk to be atypical compared to a non-*NF2* meningioma. Besides *NF2* and *SMARCB1* mutations, we did not find any other somatic mutations that significantly contributed to the risk of being atypical. Based on these findings, we included the following sentence in the main text: “These findings suggest that except for mutations in *NF2* and *SMARCB1*, coding variation does not significantly contribute to the risk of being atypical.” (new text line 167).

In our series, all of the *SMARCB1* (n=19) mutations co-occurred with a *NF2* mutation. We used Relative Risk comorbidity measure to quantify the strength of comorbidity associations between *NF2* and *SMARCB1*. We found that a *SMARCB1* mutant sample was 44 times more likely to harbor a *NF2* mutation (Fisher’s Exact test $P=1.2 \times 10^{-7}$). We revised the manuscript to include these findings (new text line 108).

8) The authors stratify samples into two groups (i.e. high/low) on the basis of the mean percentage of genome alteration (PGA). No details about the distribution of the PGA across all samples are reported. As widely accepted, the mean of a distribution is affected by outliers (or extreme values) that can cause the mean to have a higher or lower value biased in favour of the direction of the outliers. This issue can influence the subsequent analyses and, most importantly, the conclusions. The author must report and comment on the distribution of the PGA. If the mean is biased by the outliers then a tumour with a PGA close to the mean PGA can erroneously classified as CNV high/low. If the authors want to divide samples into two groups, the median is a better estimator. They must compare the mean with the median of the distribution. If the mean is significantly different from the median the author must use the median to stratify their cohort.

The reviewer raises an important point. We first plotted the distribution of PGA values (updated supplementary figure S5). The mean of PGA was 4.82% whereas the median was 1.27%. We also calculated the number of large-scale events, i.e. affecting more than 1/3 of the chromosome arm, for each sample, classifying samples with less than 2 large-scale events as “CNV-low”. As the reviewer suggested, when we used the median PGA as a cutoff, “CNV-low” class assignment of 41 samples (out of 208) changed to “CNV-high” and 29 of those samples had less than 2 large scale events. While the use of median would evenly divide our cohort into two groups, we believe that a division that keeps these relatively stable samples in the CNV-low group is more biologically relevant. For this reason, in our sample set, we hope that the reviewer would agree that mean-PGA would be a better threshold for classifying samples as “CNV-low” or “-high”.

However, in response to this suggestion, we have performed our analysis using the median as our PGA threshold. Based on the median PGA threshold, we again observed that atypical tumors had a 2.16 times greater risk to be CNV-high ($P=0.0006$ Fisher’s exact test). Similar to the previous analysis, gene expression, miRNA and DNA methylation data clustering separated CNV-high samples from CNV-low ones. When we used the median-PGA as a cutoff, sample CNV class assignments did not change for the H3K27ac and H3K27me3 data.

Updated Supplementary Figure S5: The distribution of percentage of genome alteration (PGA) across all meningioma samples. The mean (4.82%) and median (1.27%) PGA values are marked as dash lines in the plot.

9) The authors claim that atypical tumours have an mRNA expression signature that distinguishes them from benign meningiomas (p.11 line 218). This is a strong statement especially under the light of PCA analysis that clearly cannot separate atypical from benign tumours (Fig.2b and line 225). The authors present the results stratified by NF2-loss and this create confusion in interpreting results. The authors must present the results of the PCA for atypical and benign tumours without any stratification, even as a supplementary figure. The results of the PCA should be shown as first (the axes should also report the percentage of variability explained by each component). As previously done, the authors did not clearly specify in the methods any cutoff that they used for the identification of differential expressed genes. How do they define differential expressed genes? How many differentially expressed genes do they find? Apparently, a table with gene expression values in each sample is not provided. Fig 2c is not clear and, presented as it is, is not informative. Fig 2d is show as an example but it does not contain any relevant information (i.e. result of the analysis).

We appreciate the reviewer's comment. We want to clarify that the PCA plot in Figure 2b was performed based on the top 1000 genes with the highest standard deviation. We have updated the PCA plot using the mRNA signature genes (n=483). As expected, in the updated PCA plots, it can clearly be seen that this mRNA gene signature distinguished atypical from benign samples (updated Fig 2b). As requested, in the PCA plot axes, we also reported the percentage of explained variability of each principal component.

We apologize for not explicitly the cut-off values used for the identification of differential expressed genes. We identified these differentially expressed genes using an empirical Bayesian method 'eBayes' implemented in the *limma* R package¹⁵. Genes were considered to be differentially expressed with an adjusted p-value < 0.05. We revised the Methods section to specify these cutoffs used in gene expression analysis (in new text line 665) and Supplementary Table 4d list all of these differentially expressed genes. We are in the process of depositing our raw and preprocessed gene expression matrix containing all the probes across all the samples into Gene Expression Omnibus (GEO) database.

Based on the reviewer's suggestion the Figure 2c is not informative, we removed it in the revised manuscript and moved Figure 2d to supplementary figures.

Updated Supplementary Figure S6: Principal component (PC) analysis of meningioma gene expression data using mRNA signature genes separates atypical and benign samples.

Updated Supplementary Figure S7: Principal component (PC) analysis of meningioma gene expression data using mRNA signature genes separates atypical and benign samples.

Updated Supplementary Figure S8: Barplots of the percentage of variability explained by the top 10 principal components.

10) The author use a predictive algorithm (i.e. random forest) to classify meningiomas accordingly to the expression of 25 selected genes. Again, the authors do not provide any details about the implementation of this analysis in the Methods. The authors must provide these details. How do they create/train their model? Do they use the model on a test set? How do they evaluate the performance of their model? Reporting that 96% of sample was correctly identified is not sufficient. Moreover, the model is generated considering only the top 25 differentially expressed genes in atypical versus benign tumours. As a consequence, one can expect that this predictive algorithm will separate correctly the two groups. More importantly, to state that atypical tumours have a specific mRNA expression signature able to separate them form benign tumour, all differentially expressed genes should be included in the model. What are the results of this analysis if the author include all differentially expressed genes?

Thank you for your insightful comments. We have now revised Methods section to explain all the details of random forest prediction model (new text line 668). We changed the wording in the text to emphasize that 96% prediction accuracy (4%

error rate) corresponds to out of bag error (OOB) rate. We defined OOB in random forest as the error rate of the trained algorithm on a left out dataset that was not used in trained algorithm; therefore reflecting the generalized testing error of the trained model.

In the revised manuscript, we validated our prediction model on an independent meningioma gene expression dataset⁵. As we detailed above (Reviewer#2, criticism#3), in this validation cohort, we correctly predicted 42/43 benign and 6/9 atypical samples with an overall accuracy for this replication cohort of 91%.

In response to reviewer's additional comments, we also built our prediction model based on all differentially expressed genes (n=483). Our OOB error rate was 5%. When we validated this prediction model on an independent dataset, the prediction accuracy was 28%, worse than our prediction model with the top 25 genes. We believe that when we use 483 genes, the model may actually be overfitting. We also believe that with the use of the top 25 genes, we were able to capture important biological pathways underlying atypical meningiomas; for example, 15 out of these 25 genes are members of the "cell cycle" GO term (new text line 255).

11) The authors analyze DNA methylation data (p.14) but they do not comment on whether atypical tumour differs from benign ones (which should be the topics of their paper as state in p.3 line 38). It should be worth commenting on this point.

We thank the reviewer for this comment. We showed that ESC-PRC2 binding sites were hypermethylated in atypical versus benign meningiomas. Indeed, when we performed clustering analysis on DNA methylation data, we identified 4 different clusters, which clearly separated the *NF2* mutant meningiomas from relatively less methylated non-*NF2* tumors. In addition, both *NF2* and non-*NF2* meningiomas formed 2 different sub-clusters. In non-*NF2* group, atypical non-*NF2* samples clustered separately from benign non-*NF2* ones. In the *NF2* mutant group, one of the subgroups revealed a distinct hypermethylated phenotype and was significantly enriched for atypical samples, separating these from benign tumors. Based on these observations, we believe atypical tumors clearly differ from benign tumors based on genome-wide methylation data. These observations are detailed in the revised text, lines 328.

Minor comments

1) The proportion of atypical tumours in p.5 line 88 (i.e. 10%) is different from the estimates reported in p. 4 line 66 (i.e. 5-20%). Which is the real fraction of atypical meningiomas? Supplementary Table 1a and 1b must be merged since they are redundant.

We apologize that we did not make this point more clear. We revised manuscript to read "Overall, atypical tumors comprise approximately 5-20% of all meningiomas..."

in new text line 68.

2) Colour keys reported in Figures are sometimes redundant. Please reduce this redundancy. Sometimes the same colour is used in different figures to indicate distinct feature. This creates confusion. Colours must be consistent across figures. A colour must be used to indicate univocally only one feature among all figures.

We thank the reviewer for this comment. We have now reduced the color redundancy in the updated Figures 1-5.

3) At line 184 (p.9) the author rounded the pvalue present in figure 1c (0.27) to 0.2. The correct rounding is 0.3.

We thank the reviewer for this remark. We have now corrected the p-value in the revised manuscript (new text line 194).

4) The authors show that losses of chr14q, 10q, 10p, 1p and 6q are significantly different between atypical and benign meningiomas (p.10 line 195). The authors report the FDR values but they must provided also the p values of the fisher tests.

We thank the reviewer for this comment. We have now added the p-values in the revised manuscript (new text line 208).

Reviewers' Comments:

Reviewer #1 (Remarks to the Author)

While I think it is a generally interesting story, the lack of clear clinical data correlations, together with the limited novelty, is a weakness of the manuscript.

Reviewer #2 (Remarks to the Author)

The authors have satisfactorily addressed my concerns. I have no further comments.

Reviewer #3 (Remarks to the Author)

The authors answered to the points raised. I agree for publication of the revised manuscript.

We appreciate the kind comments from the reviewers. Thanks to their input, we believe the final version of the manuscript has improved.